# Genesis and Prospecting Potential of the Da'anhe Skarn Au Deposit in the Central of the Lesser Xing'an Range, NE China: Evidence from Skarn Mineralogy, Fluid Inclusions and H-O Isotopes

Chuntao Zhao [1,2,3], Fanting Sun [3,*], Jinggui Sun [3], Jianping Wang [1,2,*], Jilong Han [3,4], Xiaolei Chu [3], Chenglin Bai [3,5], Dongmei Yu [1,2], Zhikai Xu [3], Lei Yi [1,2] and Shan Hua [6]

[1] Key Laboratory of Green and High-End Utilization of Salt Lake Resources, Qinghai Institute of Salt Lakes, Chinese Academy of Sciences, Xining 810008, China; zhaoctao22@isl.ac.cn (C.Z.); ydm2011@isl.ac.cn (D.Y.); yilei@isl.ac.cn (L.Y.)

[2] Qinghai Provincial Key Laboratory of Geology and Environment of Salt Lakes, Xining 810008, China

[3] College of Earth Science, Jilin University, Changchun 130061, China; sunjinggui@jlu.edu.cn (J.S.); hanjilong@cugb.edu.cn (J.H.); chuxiaolei19@163.com (X.C.); 3057220016@email.cugb.edu.cn (C.B.); xuzk@qdio.ac.cn (Z.X.)

[4] Development and Research Center of China Geological Survey, Beijing 100037, China

[5] School of Earth Sciences and Resources, China University of Geosciences, Beijing 100083, China

[6] The Ninth Geological Brigade, Hebei Bureau of Geology and Mineral Resources Exploration, Xingtai 054000, China; m15903218986@163.com

* Correspondence: sunft21@mails.jlu.edu.cn (F.S.); wang_ktz2022@126.com (J.W.)

**Abstract:** Skarn Au deposits exist in the circum-pacific metallogenic belt. Interestingly, the Da'anhe Au deposit is the only independent skarn gold deposit in the Lesser Xing'an Range. To determine the metallogenic mechanism and prospecting potential of the Da'anhe deposit, we performed skarn mineralogy, fluid inclusion (FI) and H-O isotope analyses. The results show the following: (1) The Da'anhe deposit is a calcareous reduced skarn Au deposit that formed between an Early Jurassic gabbroic diorite and the Permian Tumenling Formation marble. Its metallogenic process includes five stages: the early skarn stage (Stage $I_1$), late skarn stage (Stage $I_2$), early quartz-sulfide stage (Stage $II_1$), late quartz-sulfide stage (Stage $II_2$) and quartz-carbonate stage (Stage $II_3$). Gold precipitated in Stage $II_1$ and Stage $II_2$. (2) The initial ore-forming fluid was derived from magmatic water and featured a high temperature and intermediate to high salinity. After boiling and mixing, the fluid eventually changed to a low-temperature and low-salinity reducing fluid dominated by meteoric water. (3) The formation depth of the Au orebodies was 2.27–3.11 km, and the orebodies were later lifted to the surface (<500 m). The potential for finding skarn Au deposits in the study area is limited. (4) The distinctive nature of the ore-related magma (i.e., source, reducing conditions and high water content) was key to the formation of the Da'anhe skarn gold deposit.

**Keywords:** mineralization; skarn mineralogy; H-O isotopes; fluid inclusion; Da'anhe Au deposit; lesser Xing'an range

## 1. Introduction

Skarn deposits are important sources of iron, copper, lead, zinc, molybdenum, gold, tungsten and other metal minerals worldwide. In the circum-pacific metallogenic belt, gold production from skarn deposits has exceeded 1000 tons [1,2]. Skarn gold deposits in China are distributed in various tectonic provinces and are hosted in rocks ranging in age from the Palaeoproterozoic to the Cretaceous. At least seventy deposits have been identified as gold or gold-bearing skarn deposits in China [3]. Northeastern China, located in the western circum-pacific metallogenic belt, was affected by the subduction of the

Paleo-Pacific plate and extension after the collision and orogeny between the Laurasian and Gondwanan continents. The Lesser Xing'an Range in northeastern China experienced extensive magmatism and mineralization during the Palaeozoic and Mesozoic, particularly the latter which was an important epoch for numerous skarn polymetallic deposits [4–6]. Strong tectonic-magmatic activity provided favourable conditions for the formation of skarn Au deposits in this region [7,8].

However, most of these deposits are calcareous skarn deposits that are composed mainly of iron, copper lead and zinc and are genetically related to the metasomatism of intermediate-acidic plutonic rocks and marble. Examples of such deposits include the Ergu Fe-Zn polymetallic deposit [9–12], Cuihongshan Fe-Cu-W-Mo polymetallic deposit [13–15], and Xulaojiugou Pb-Zn polymetallic deposit [16,17]. The medium-sized Da'anhe gold deposit is the only skarn Au deposit. At present, several geologists have performed detailed studies on the metallogenic mechanisms and processes of the skarn Fe-Cu-Pb-Zn deposits in this region [13,17,18], but research on the Da'anhe Au deposit has focused mainly on geological investigations and the chronologic and geochemical characteristics of plutonic rocks [19–21]. Identifying the genetic and mineralization processes of the Da'anhe deposit, and comparing them with those of the skarn Fe-Cu-Pb-Zn deposit in this region would be beneficial and is necessary for researching the metallogenic specificity of skarn deposits. Therefore, we recently carried out detailed field geological investigations, skarn mineralogy analysis, fluid inclusion (FI) analysis, coupled with H-O isotope research on the deposits, aiming to further determine the metallogenic mechanism and metallogenic process and provide a reference for prospecting for skarn Au deposits in the study area.

## 2. Regional Geological Characteristics

The Da'anhe deposit is located in the eastern section of the Xingmeng orogenic belt and is sandwiched between the Siberian, North China and Paleo-Pacific plates. This superimposed tectonic regime has experienced the evolution of the Palaeo-Asian Ocean tectonic system and the Xingmeng orogeny in the Palaeozoic and the subduction of the Paleo-Pacific plate and Okhotsk Ocean plate in the Mesozoic [22–25] (Figure 1A,B) [26,27]. The multi-stage tectonic-magmatic activities resulted in the Lesser Xing'an Range being a typical basin-mountain region with endogenetic metal development related to tectonic-magmatic activity [28–30]. The stratigraphy in this region is dominated by the silicoferrite of the Dongfengshan Formation (821–801 Ma [31]) and overlying Cambrian–Ordovician neritic facies sedimentary rocks (limestone, dolomite marble, carbonaceous slate, etc.; ~500 Ma [32]), Permian terrigenous clastic-carbonate rocks (amphibolite, marble, tuff, felsic schist, slate, etc.; 262–286 Ma [10]), Mesozoic continental volcanic rocks, and Palaeogene fluvial and lacustrine clastic sedimentary rocks. The study area experienced at least five stages of magmatism: middle Cambrian-Late Ordovician (508~450 Ma [30]), middle-late Permian (272~260 Ma [29]), Early-Middle Triassic (252~234 Ma [29]), Late Triassic-Early Jurassic (225~197 Ma [28,29,33]) and Late Jurassic (191~175 Ma [34,35]). Voluminous Phanerozoic granitoids, including abundant I-type granites and a few A- and S-type granites are present in this region [36]. In addition to these granitoids, rare mafic-ultramafic intrusions are also present in the region. Among them, Late Triassic and Early Jurassic granites are the most developed and widely distributed. The large-scale mineralization of porphyry molybdenum and skarn deposits mainly occurred at 186~176 Ma [24], and epithermal gold mineralization mainly occurred at 110~95 Ma [24]. Many porphyry molybdenum deposits in different scales (Luming, Huojihe, Cuiling, and Gaogangshan) formed in the Mesozoic granitic complex (Figure 1C) [28]. Many skarn deposits (such as the Cuihongshan, Ergu and Xulaojiugou deposits) formed in the contact belts between Mesozoic granite and Palaeozoic carbonate strata (Figure 1C) [28]. Most epithermal deposits are located in the Mesozoic to Cenozoic basins that are composed of continental volcanic and sedimentary rocks, such as the Dong'an, Gaosongshan, Yongxin, Tuanjiegou, and Yidonglinchang deposits. In this study, the Da'anhe gold deposit is essentially consistent with the mineralization of the widely developed skarn polymetallic deposits and porphyry molybdenum deposits,

which formed during the late Triassic to Early-Middle Jurassic (203.4~179.9 Ma) [10,16,28] (Figure 1C) [28].

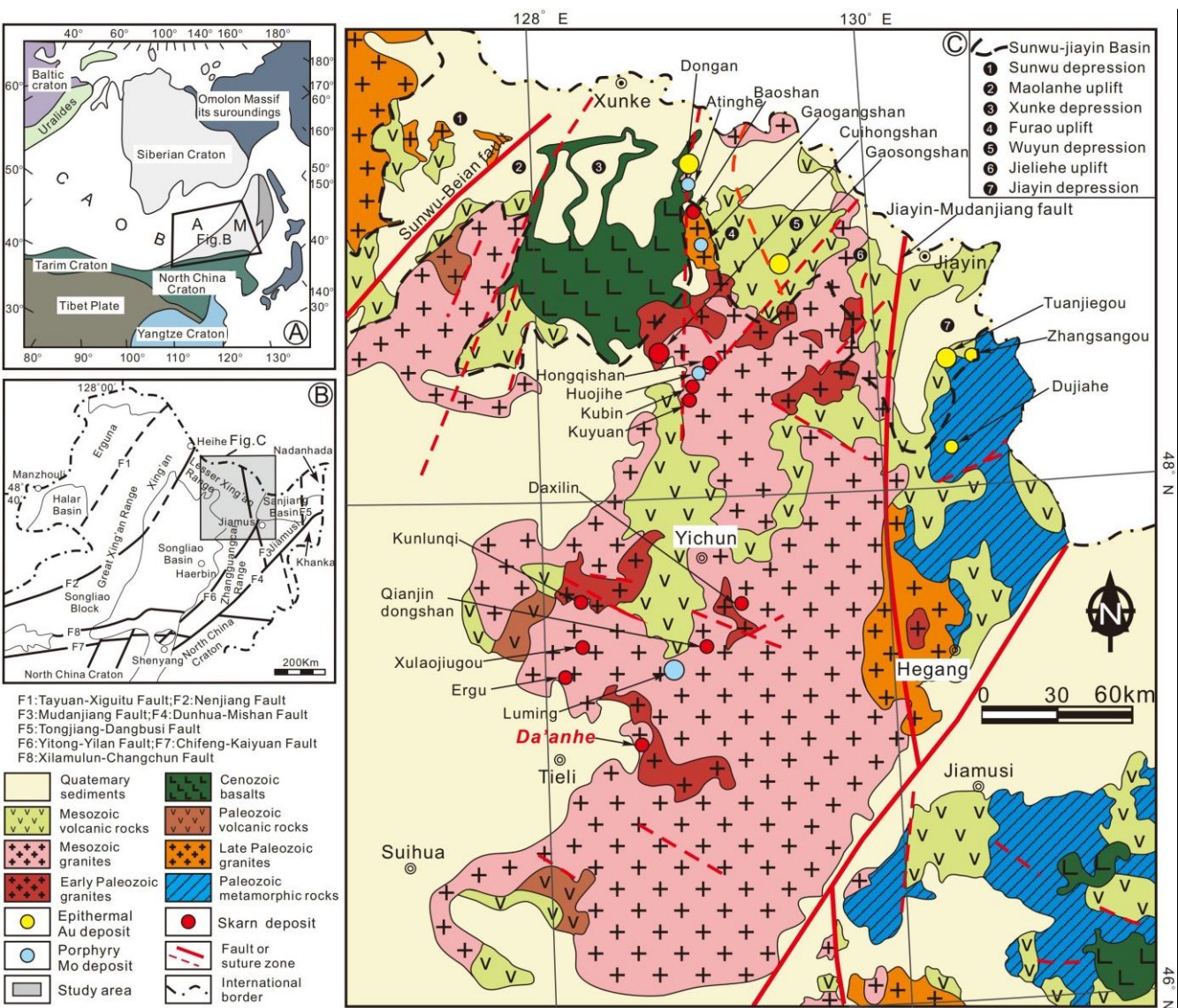

**Figure 1.** (**A**) Location of Northeast China with respect to the main tectonic units of China and Russia, modified after Wu et al. [26]. 'A' and 'M' represent the Altaids and Manchurids, respectively. (**B**) Tectonic sketch maps of Northeast China, modified after Wu et al. [27]. (**C**) Simplified geological map of the Lesser Xing'an range, modified after Wu et al. [28].

## 3. Deposit Geology

### 3.1. Stratigraphy, Magmatism and Structure

The Da'anhe deposit is located approximately 30 km northeast of the Tieli city, Heilongjiang Province, and is tectonically part of the southern Yichun-Yanshou magmatically and tectonically active zone [19–21]. The strata in the mining area include the Permian Tumenling Formation and the Jurassic Tai'antun Formation (Figure 2) [21]. The Tumenling Formation is distributed in the Sanjiaoshan area of the mining area in a discontinuous belt trending in the NE direction. The rock types include slate, metamorphic quartz sandstone, marble and andesitic crystal tuff. Marble is distributed in metamorphic quartz sandstone strata with lenticular or irregular shapes and surrounds Au ore bodies. The Tai'antun Formation is less exposed in the mining area and is distributed in the northeastern part of

the mining area in the form of residual bodies. The rock types are mainly andesitic tuff and andesite, which unconformably overlie the Tumenling Formation.

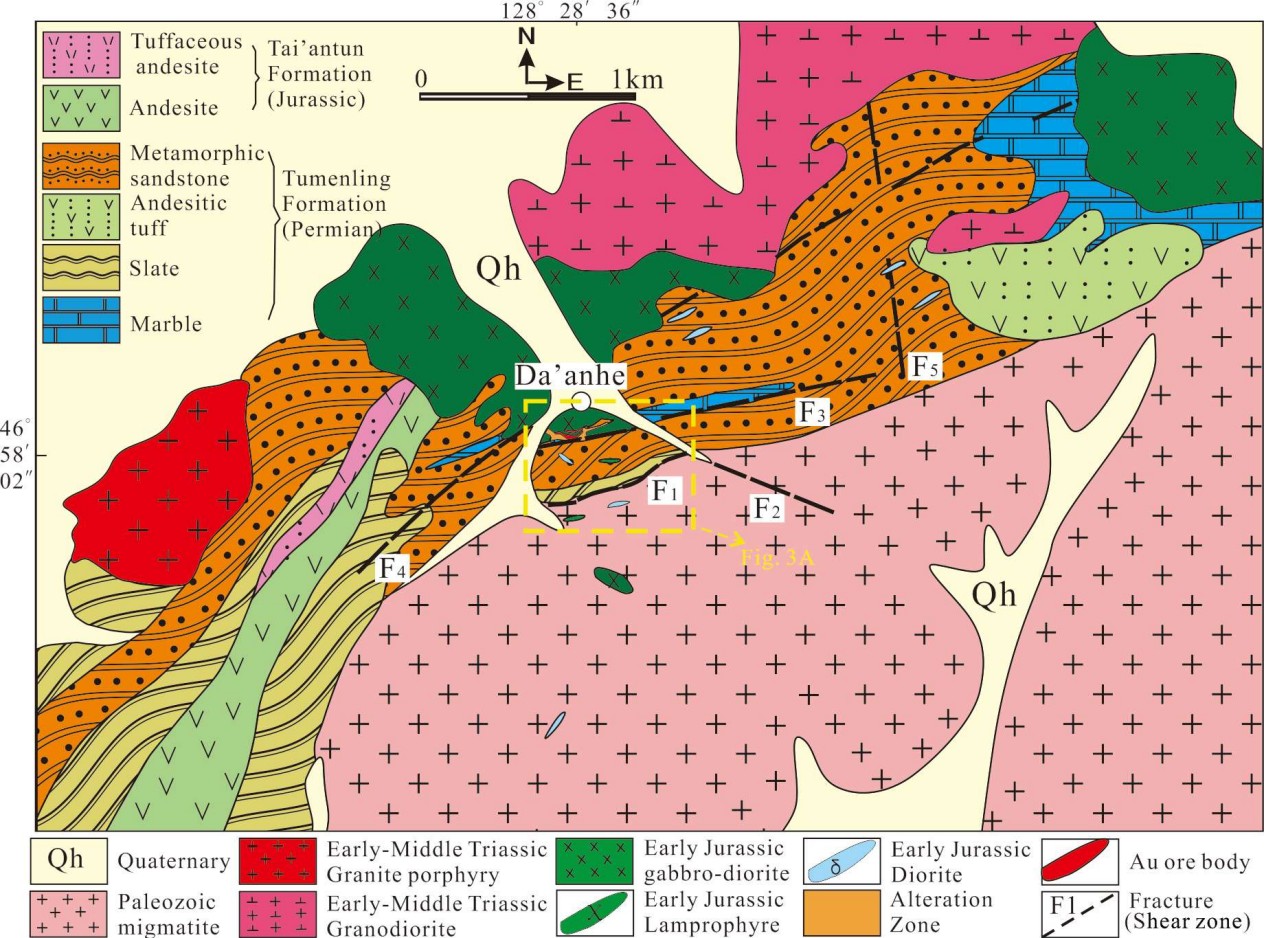

**Figure 2.** Geological map of the Da'anhe Au deposit (Figure 2 modified according to Yang [21]. Figure 3A detailed in Figure 3).

The magmatic rocks in the mining area are mainly Palaeozoic migmatite granite and Mesozoic granodiorite, granodiorite porphyry and gabbroic diorite (Figure 2) [21]. Palaeozoic migmatite is exposed in the southern mining area in the form of a large-scale batholith. The granodiorite and granodiorite porphyry formed in the Early-Middle Triassic and were emplaced in the form of stocks in the Tumenling Formation along the NE-trending Jinniugou-Da'anhedongshan fault in the northern part of the mining area. The gabbroic diorite on both sides of the ore body has undergone significant alteration (see Section 3.2.1) and formed endoskarn, which occurs in the inner belt of the skarn, indicating a close relationship with mineralization. In addition, late lamprophyre and fine-grained diorite dikes are present in the mining area.

The structures in the mining area are mainly faults and folds. The faults in the mining area mainly include the NE-trending Jinniugou-Da'anhedongshan fault, the NW-trending Da'anhe fault and the nearly NW-trending Da'anhenanshan fault. These faults are mainly manifested as breccia zones, shear zones and schistosity zones in the mining area. The Jinniugou-Da'anhedongshan fault and Da'anhe fault intersect and become the main ore controlling structures in the mining area. The large fold in the region is the Shenshu complex anticline, and the mining area and the deposit are located on the southeastern limb of the NE-striking Sanjiaoshan syncline within the Shenshu anticlinorium (Figure 2) [21].

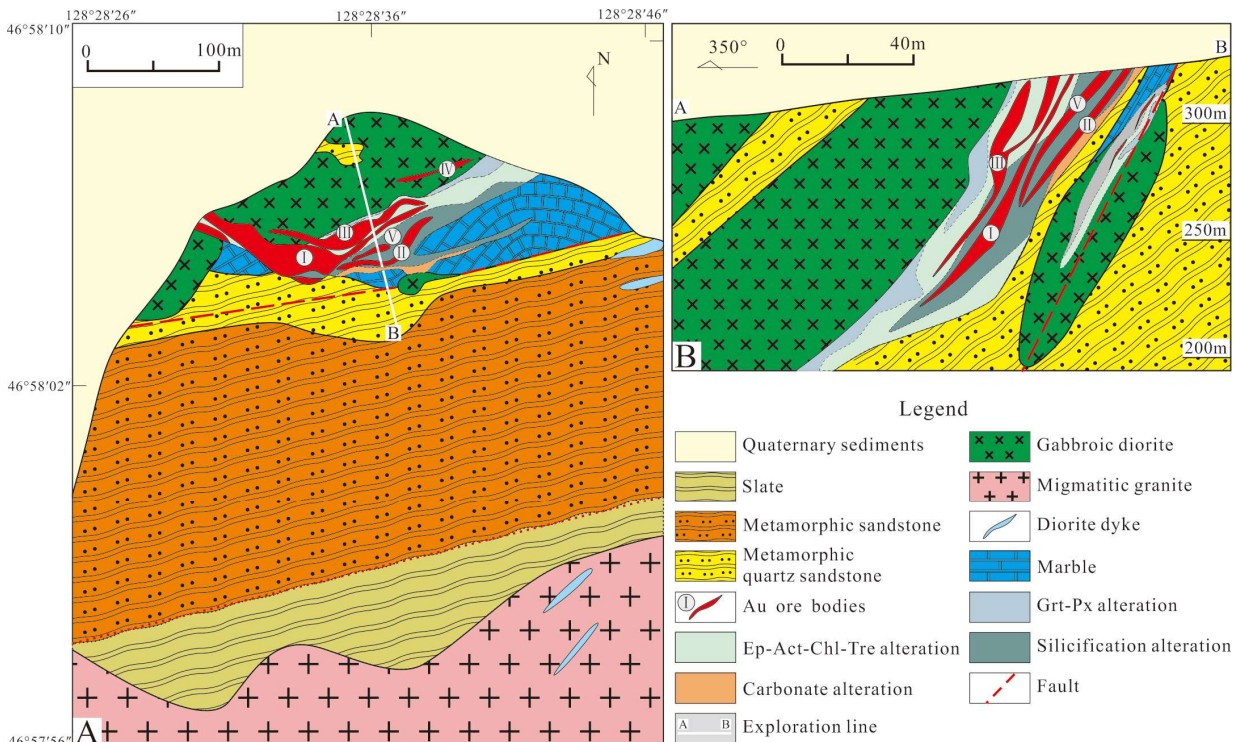

**Figure 3.** (**A**) Geological map of the Da'anhe Au deposit; (**B**) Profile of exploration line 4 of the Da'anhe Au deposit. ((**A,B**) modified according to Xue [20] and Yang [21]). Note: F1 and F3: Da'anhenanshan fault; F2: Da'anhe fault; F4: Jinniugou-Da'anhedongshan fault. Grt-garnet; Px-diopside; Ep-epidote; Act-actinolite; Tr-tremolite; Chl-chlorite.

### 3.2. Mineralization

The gold orebodies in the Da'anhe deposit mainly developed in the skarn belt between the gabbroic diorite (Figures 3, 4A and 5A) and the marble of the Tumenling Formation in the central and western mining areas, and small amounts of altered gabbroic diorite and marble developed on both sides of the skarn belt. The orebodies are controlled by shear zones and occur in NE- or NW-striking lenses or veins dipping steeply between 42 and 87 degrees. The ore bodies are generally approximately 25 m long, and some ore bodies can reach lengths of up to 168 m. At present, 12 industrial orebodies have been identified in the Da'anhe deposit, and the main orebodies are the No. I, II and III orebodies. The three main orebodies occur as veins in the inner and outer belts of the skarn zone, and the average grades of gold are 11.93 g/t, 46.78 g/t and 5.93 g/t, respectively.

#### 3.2.1. Alteration of the Surrounding Rock

The wallrock alteration in the mining area is relatively well developed. The main alteration types are Grt-Px alteration (Figures 3 and 4B–D), Ep-Act-Chl-Tr alteration (Figures 3 and 4E,F), sericitization (Figure 4D), silicification and carbonatization (Figures 3 and 4L). Among them, Grt-Px alteration and silicification are most closely related to gold mineralization. Grt-Px alteration occurs mainly between the gabbroic diorite and the marble of the Tumenling Formation. Silicification occurs mainly in the outer belt of the skarn, and silicified veins of different colours and sizes cut through the early skarn minerals and sulfide veins. Carbonatization occurs in the late stage of mineralization and occurs in the tectonically active part in the form of veins. In addition, potassic alteration and tremolite alteration occur sporadically in the mining area.

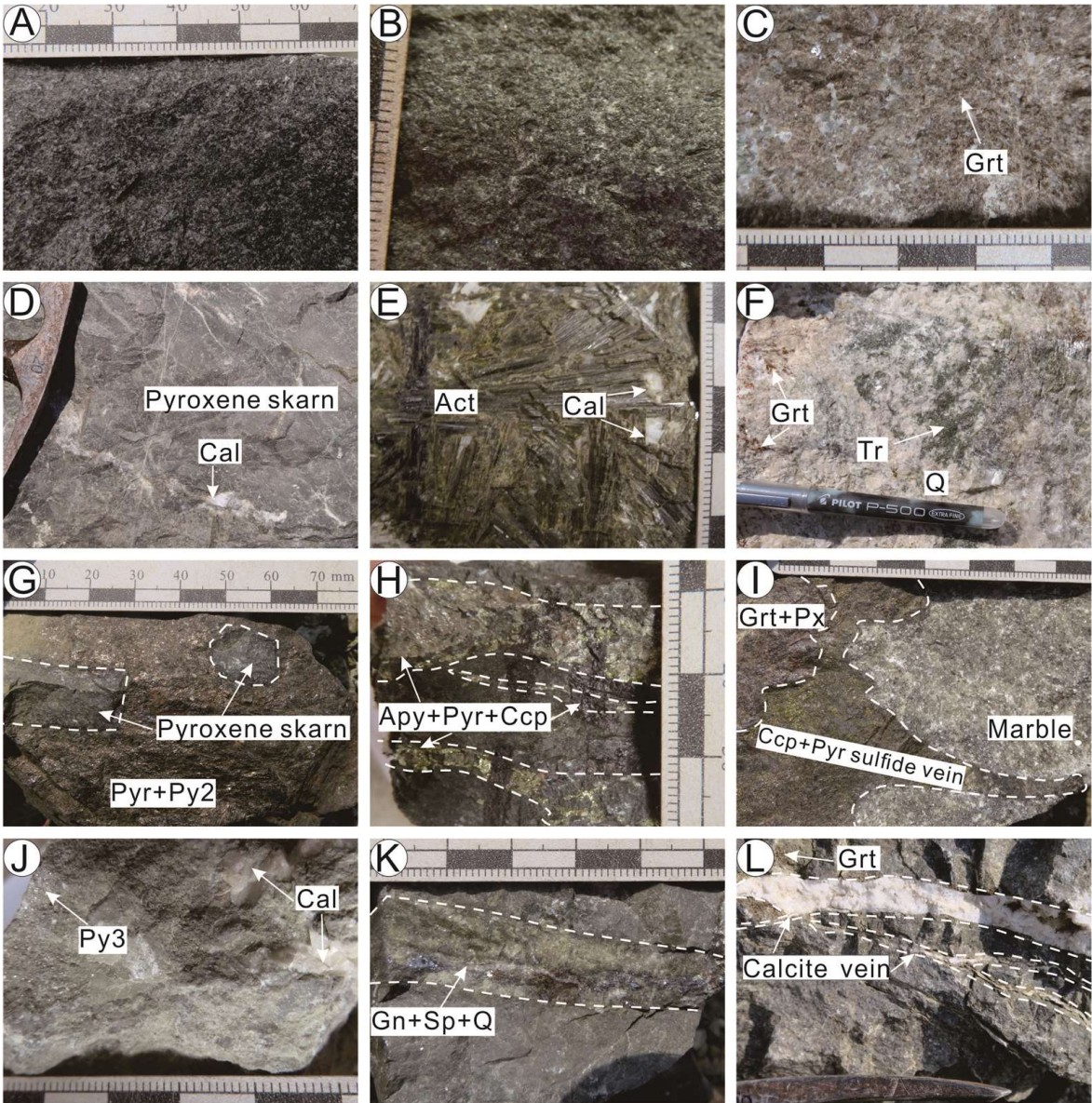

**Figure 4.** Field geological characteristics of the Da'anhe gold deposit. (**A**). Unaltered gabbroic diorite; (**B**). Skarnized gabbroic diorite hand specimen photographs; (**C**). Photographs of garnet skarn hand specimens; (**D**). Diopside skarn was cut through by late carbonate veins; (**E**). Photographs of skarn hand specimens from actinolite skarn; (**F**). The garnet in skarn metasomatized by late tremolite and quartz began to appear; (**G**). The pyroxene skarn formed in the dry skarn stage was broken into a breccia shape under the influence of tectonism and cemented by late sulfide; (**H**). Sulfide vein penetrating skarn containing arsenopyrite pyrrhotite chalcopyrite; (**I**). Sulfide veins containing chalcopyrite pyrrhotite cut through garnet pyroxene skarn and marble; (**J**). Granular pyrite ore with good crystallinity cut through by late carbonate vein; (**K**). Quartz vein containing galena sphalerite; (**L**). The carbonate veins of the late mineralization cut through the early garnet skarn. Note: Grt-garnet; Px-diopside;Act-actinolite; Tr-tremolite; Q-quartz; Cal-calcite; Ccp-chalcopyrite; Apy-arsenopyrite; Pyr-pyrite; Py2-pyrite coexisting with quartz; Py3-pyrite coexisting with calcite; Gn-galena; Sp-sphalerite.

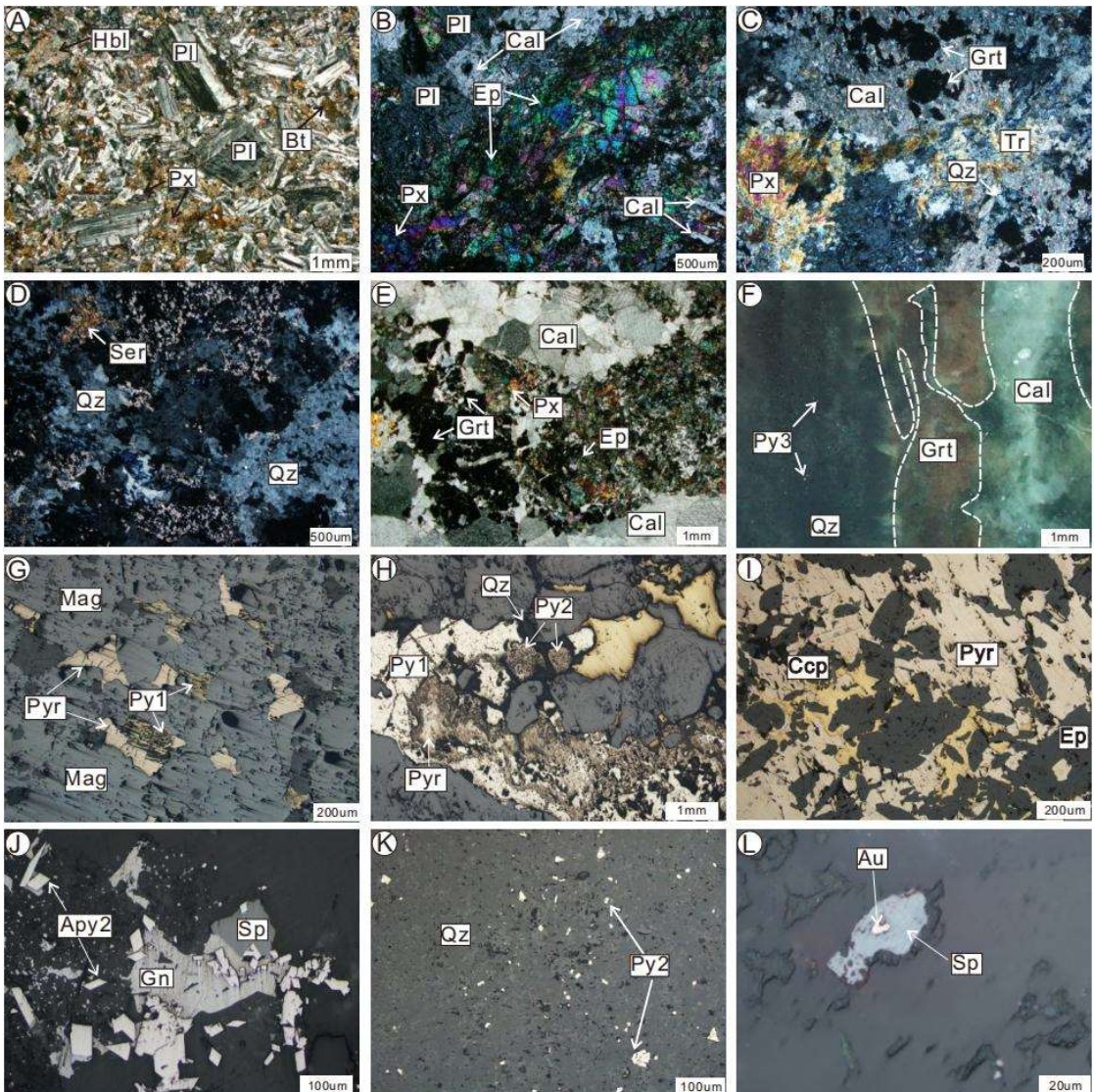

**Figure 5.** Microscopic characteristics of rocks and ores in the Da'anhe deposit (cross-polarized light). (**A**). Microphotograph of gabbroic diorite; (**B**). Microscopic characteristics of gabbroic diorite in skarn alteration; (**C**). Garnet and pyroxene were metasomatized by quartz and tremolite and were cut through by the late carbonate vein; (**D**). Sericitization alteration; (**E**). Skarn alteration of marble; (**F**). Garnet was replaced by pyrite bearing quartz veinlets and calcite veinlets in later stage; (**G**). Magnetite shows its granular structure after replacement by pyrite and pyrrhotite; (**H**). The early pyrite and pyrrhotite were replaced by the later pyrite and brass bearing quartz veins; (**I**). Chalcopyrite, pyrrhotite and epidote coexist; (**J**). Sphalerite and galena coexist, and metasomatic arsenopyrite; (**K**). Semiautomorphic pyrite particles in late mineralization; (**L**). Encapsulated gold developed in sphalerite. Note: Hbl-hornblende; Pl-plagioclase; Px-pyroxene; Bt-biotite; Ep-epidote; Cal-calcite; Grt- garnet; Tr- tremolite; Qz-quartz; Ser-sericite; Pyr-pyrite; Py1-early pyrite; Py2-pyrite coexisting with quartz; Py3-pyrite coexisting with calcite; Mag-magnetite; Ccp-chalcopyrite; Apy2-late arsenopyrite; Au-natural gold; Gn-galena; Sp-sphalerite.

### 3.2.2. Characteristics and Stages of Mineralization

The ore minerals in the ores are widely distributed and mainly include arsenopyrite (Figure 4H), pyrrhotite (Figure 4G–I), chalcopyrite (Figure 4H,I), pyrite (Figure 4G,J), sphalerite (Figure 4K), native gold, electrum, and small amounts of magnetite, galena (Figure 4K), malachite and bismuth telluride. Among them, gold occurs mainly in the form of native gold or electrum in skarn minerals, quartz or sulfide grains. The ore occurs in

banded (Figure 4H), vein (Figure 4I,K) and sparse disseminated (Figure 4J) forms, followed by dense massive and brecciated (Figure 4G) forms. According to the cross-cutting relationships of the veins and the mineral assemblage characteristics under the microscope, the metallogenic process of the Da'anhe deposit can be divided into askarn period and a quartz-sulfide period, with a total of five stages. (1) In the early skarn stage (Stage $I_1$), high-temperature anhydrous skarn minerals, such as garnet and diopside minerals, formed (Figure 5B,C). (2) The late skarn stage (Stage $I_2$) did not have a major impact on the mining area and mainly formed water-bearing skarn minerals such as actinolite, tremolite and epidote (Figure 5C,E), and a small amount of magnetite (Figure 5G). (3) In the early quartz-sulfide stage (Stage $II_1$), skarn minerals were largely metasomatized, and gangue minerals, such as quartz, epidote, chlorite and sericite, were formed (Figure 5C,D). In addition to arsenopyrite, the ore minerals were mainly iron-copper sulfides, including pyrrhotite, pyrite and chalcopyrite (Figure 5G–K), and a small amount of native gold precipitation began in the late part of this stage. (4) The late quartz-sulfide stage (Stage $II_2$) was the main metallogenic stage of gold. Gold precipitated as native gold or electrum between skarn minerals and quartz or metal sulfide grains (Figure 5L). In this stage, the gangue minerals are mainly quartz and calcite, and the ore minerals are mainly pyrite, galena and sphalerite (Figure 5J,K). (5) In the quartz carbonate stage (Stage $II_3$), only small amounts of pyrite, sphalerite and other sulfides formed, and many quartz-calcite veins formed, cutting through the early skarn minerals or quartz sulfide veins (Figure 5F), signifying the end of mineralization [20,21].

## 4. Samples and Analytical Methods

### 4.1. Electron Probe Analysis

Initially, garnet-pyroxene skarn samples collected from the surface of the Da'anhe deposit were cut into pieces for microprobe analysis. On the basis of detailed microscopic identification, garnet and pyroxene were selected and analysed by an electron probe (EPMA-1600, Japan Shimadzu Company, Kyoto, Japan) at the Jilin University Testing and Analysis Center. Standard samples of 53 kinds of minerals were obtained from the SPI Company, Lakewood, WA, USA. The element analysis range was 5B-92U, the electron beam current was stable at $1.5 \times 10^{-3}$/h, the accelerating voltage was 15 kV, the current was 4.5 nA, the beam spot was less than 1 μm, and the correction method was ZAF. Geokit software (GeokitPro version) was used to calculate the occupation and endmember composition of each element in the mineral [37].

### 4.2. Temperature Measurements and Laser Raman Analysis of Fluid Inclusions (FIs)

The analysed samples consisted of garnet and pyroxene from the early skarn stage, epidote from the late skarn stage, quartz ($Q_1$) from the early quartz-sulfide stage, quartz ($Q_2$) from the late quartz-sulfide stage and calcite from the quartz-carbonate stage. Samples for the FI study were collected from the surface of the Da'anhe deposit. For the selected samples, first, FI thin sections (thickness ~0.3 mm) were prepared, and then the type, distribution, gas-liquid ratio and daughter minerals of Fis were carefully identified under a high-magnification microscope. From the above observations, the primary inclusions were selected for the temperature measurement experiment, and the analysis was performed in the FI laboratory at the Institute of Geology and Geophysics of the Chinese Academy of Sciences. The experiment was carried out on a THMSG 600 (Linkam Company, Salfords, UK) heating and cooling stage. The measurable temperature ranged from −196 to 600 °C, and the temperature increase and decrease rate was approximately 3 °C/min. Near the phase transition temperature, the rate was maintained at 0.2~0.4 °C/min. The measurement errors of the homogenization temperature and freezing point temperature were ±2 °C and ±0.1 °C, respectively. The analysis procedure is described in detail in Fan et al. [38]. The salinity of the salt solution was calculated using the freezing point temperature [39].

To systematically determine the fluid composition during the mineralization process, representative single FIs captured during each stage were selected for gas phase composi-

tion analysis under nondestructive conditions. The experimental instrument used was a LahKam laser confocal Raman spectrometer (produced by Horiba Jobin Yvon, Montpellier, France), and the excitation wavelength was 532 nm. The test was performed by the authors in the FI laboratory at the Institute of Geology and Geophysics, Chinese Academy of Sciences.

*4.3. H-O Isotope Analysis*

The samples used for H-O isotope testing were garnet (2 samples, Stage $I_1$), epidote (1 sample, Stage $I_2$), quartz (1 sample, Stage $II_2$) and calcite (1 sample, Stage $II_3$). Samples for H-O isotope analysis were collected from the surface of the Da'anhe deposit. Initially, each selected sample was crushed and ground to 40–80 mesh, and a pure sample with a purity greater than 99% was selected for analysis under a binocular microscope. The samples were tested after washing and desorbing water and secondary FIs. The test was completed at the Beijing Experimental Center of Nuclear Industry. The experimental analysis instrument was a Finnigan-MAT253 mass spectrometer. Oxygen isotope analysis was carried out using the BrF5 method. Mineral oxygen was extracted by the reaction of BrF5 with oxygen-containing minerals under vacuum and high-temperature conditions and converted into $CO_2$ gas by burning with a thermally resistant graphite rod. The analytical accuracy was $\pm 0.2$‰. The H isotopes of the FIs were analysed by the zinc method with an accuracy of $\pm 2$‰. The international standard for H and O isotopes was SMOW.

To calculate the $\delta^{18}O_{V\text{-}SMOW}$ values of fluids in equilibrium with silicate minerals, the following fractionation equations were used: the garnet-water equation of Taylor [40], the epidote-water equation of Zheng [41], the quartz-water equation of Clayton et al. [42], and the calcite-water equation of Zheng [41].

**5. Results**

*5.1. Electron Probe Analytical Results for the Skarn Minerals*

5.1.1. Garnet

Only one type of garnet has been found in the Da'anhe mining area. In hand specimens, the garnet is light red to reddish-brown with particle sizes of 0.3~1.5 mm, and it is often associated with pyroxene and replaced by later minerals (Figure 6A). The garnet is homogeneous under orthogonal polarization, shows total extinction and a metasomatic residual structure, and exhibits a heteromorphic anhedral morphology. The results of the electron probe analysis of the garnet are listed in Table 1. The results show that the main components of the garnet are $SiO_2$ (average = 37.84%), CaO (average = 35.56%), FeO (average = 12.13%) and $Al_2O_3$ (average = 12.20%), with small amounts of $TiO_2$, MnO, MgO and trace $Cr_2O_3$. The garnet in the Da'anhe deposit is grossular ($Gr_{44.93-74.76}Ad_{21.89-52.89}$), with small amounts of andradite, uvarovite, pyrope and spessartite. On the garnet endmember discrimination diagram, the garnet plots within the range of typical skarn Au deposits worldwide (Figure 6C) [1].

5.1.2. Pyroxene

Pyroxene is another important skarn mineral in the Da'anhe deposit that is associated with garnet and is mostly replaced by silicate minerals, quartz, calcite and other late water-bearing minerals. There is a single type of pyroxene, and the pyroxene is usually dark green in hand specimens. Bright interference colours are observed under orthogonally polarized light. The minerals exhibit high protrusions, surface cracks, and euhedral and subhedral granular textures (Figure 6B) and the particle sizes are generally 0.5~2 mm under a polarization microscope. The electron probe analytical results for pyroxene are listed in Table 2. The analytical results show that pyroxene is mainly composed of $SiO_2$ (average = 54.34%), CaO (average = 25.20%), MgO (average = 11.60%) and FeO (average = 9.05%); small amounts of $Al_2O_3$, MnO and $Na_2O$; and trace amounts of $TiO_2$, $Cr_2O_3$ and $K_2O$ (Table 2). The pyroxene in the Da'anhe deposit is diopside rich ($Di_{65.31-70.25}Hd_{28.67-33.65}$), with small amounts of hedenbergite and johannsenite, belonging to the diopside-hedenbergite solid solution

series. In addition, on the discrimination diagram of pyroxene endmembers, the pyroxene plots in the range of typical skarn Au deposits worldwide (Figure 6D).

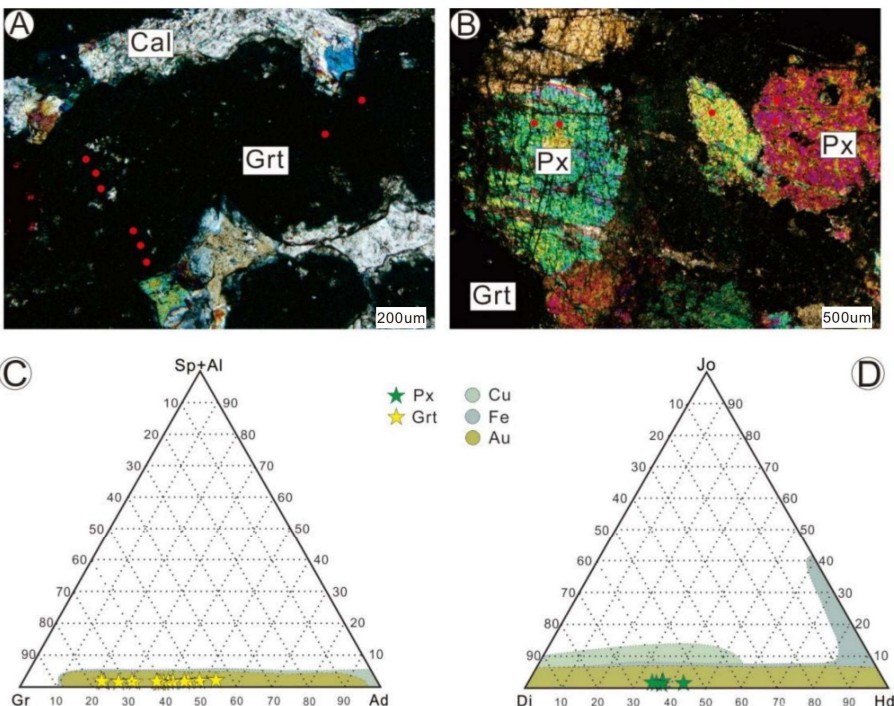

**Figure 6.** (**A,B**): Skarn mineral microscopic characteristics of the Da'anhe gold deposit (cross-polarized light); (**C,D**): Composition of garnet (data taken from Table 1) and pyroxene (data taken from Table 2) from gold skarn deposits in the study area and comparison with garnet and pyroxene from other types of skarn deposits in the world [1]. The red dot represents the testing location.

**Table 1.** Electron microprobe analyses of garnet from the Da'anhe deposits.

| Sample | Unit | 1-1 | 1-2 | 1-3 | 1-4 | 2-1 | 2-2 | 2-3 | 2-4 | 1-4-1 | 1-4-2 | 1-4-3 | 1-4-4 | 2-1-1 | 2-1-2 | 2-1-3 | 2-1-4 |
|---|---|---|---|---|---|---|---|---|---|---|---|---|---|---|---|---|---|
| SiO$_2$ | wt.% | 37.60 | 38.45 | 38.10 | 37.29 | 36.85 | 36.84 | 37.82 | 37.68 | 38.09 | 37.89 | 38.05 | 39.06 | 38.14 | 38.48 | 38.36 | 38.14 |
| TiO$_2$ | wt.% | 0.79 | 0.56 | 0.64 | 0.76 | 1.01 | 0.69 | 0.94 | 0.70 | 0.59 | 0.35 | 0.96 | 0.12 | 1.01 | 0.80 | 0.87 | 0.91 |
| Al$_2$O$_3$ | wt.% | 11.25 | 13.24 | 12.79 | 11.24 | 9.10 | 9.72 | 11.99 | 12.48 | 13.21 | 11.76 | 12.55 | 14.98 | 14.35 | 16.02 | 15.79 | 15.87 |
| Cr$_2$O$_3$ | wt.% | 0.07 | 0.00 | 0.09 | 0.04 | 0.17 | 0.01 | 0.00 | 0.06 | 0.00 | 0.00 | 0.00 | 0.00 | 0.18 | 0.12 | 0.18 | 0.21 |
| FeO | wt.% | 13.51 | 10.50 | 11.50 | 12.88 | 15.75 | 14.98 | 12.35 | 11.57 | 11.44 | 13.60 | 12.08 | 9.41 | 8.15 | 6.94 | 6.97 | 6.83 |
| MnO | wt.% | 0.19 | 0.16 | 0.22 | 0.19 | 0.33 | 0.28 | 0.16 | 0.29 | 0.12 | 0.26 | 0.11 | 0.25 | 0.17 | 0.19 | 0.17 | 0.15 |
| MgO | wt.% | 0.18 | 0.14 | 0.11 | 0.13 | 0.23 | 0.16 | 0.21 | 0.13 | 0.07 | 0.09 | 0.11 | 0.03 | 0.59 | 0.60 | 0.56 | 0.68 |
| CaO | wt.% | 35.47 | 36.04 | 35.82 | 35.95 | 34.30 | 35.41 | 35.64 | 35.89 | 35.78 | 35.11 | 36.00 | 35.84 | 35.07 | 35.48 | 35.80 | 35.45 |
| Na$_2$O | wt.% | 0.10 | 0.05 | 0.02 | 0.04 | 0.02 | 0.07 | 0.05 | 0.03 | 0.00 | 0.02 | 0.02 | 0.04 | 0.10 | 0.08 | 0.06 | 0.05 |
| K$_2$O | wt.% | 0.02 | 0.03 | 0.01 | 0.00 | 0.00 | 0.03 | 0.00 | 0.01 | 0.01 | 0.00 | 0.01 | 0.00 | 0.03 | 0.12 | 0.00 | 0.04 |
| Total | wt.% | 99.17 | 99.15 | 99.28 | 98.52 | 97.76 | 98.20 | 99.21 | 98.78 | 99.31 | 99.08 | 99.88 | 99.74 | 97.77 | 98.82 | 98.75 | 98.33 |
| **Based on 12 oxygen atoms** | | | | | | | | | | | | | | | | | |
| Si | apfu | 2.97 | 3.01 | 2.99 | 2.97 | 2.98 | 2.96 | 2.98 | 2.98 | 2.98 | 2.99 | 2.97 | 3.02 | 3.01 | 2.99 | 2.98 | 2.98 |
| Ti | apfu | 0.05 | 0.03 | 0.04 | 0.05 | 0.06 | 0.04 | 0.06 | 0.04 | 0.03 | 0.02 | 0.06 | 0.01 | 0.06 | 0.05 | 0.05 | 0.05 |
| Al | apfu | 1.05 | 1.22 | 1.18 | 1.06 | 0.87 | 0.92 | 1.11 | 1.16 | 1.22 | 1.09 | 1.15 | 1.36 | 1.33 | 1.47 | 1.45 | 1.46 |
| Cr | apfu | 0.00 | 0.00 | 0.01 | 0.00 | 0.01 | 0.00 | 0.00 | 0.00 | 0.00 | 0.00 | 0.00 | 0.01 | 0.01 | 0.01 | 0.01 | 0.01 |
| Fe3+ | apfu | 0.89 | 0.69 | 0.75 | 0.86 | 1.06 | 1.01 | 0.81 | 0.76 | 0.75 | 0.90 | 0.79 | 0.61 | 0.54 | 0.45 | 0.45 | 0.45 |
| Fe2+ | apfu | 0.00 | 0.00 | 0.00 | 0.00 | 0.00 | 0.00 | 0.00 | 0.00 | 0.00 | 0.00 | 0.00 | 0.00 | 0.00 | 0.00 | 0.00 | 0.00 |
| Mn | apfu | 0.01 | 0.01 | 0.01 | 0.01 | 0.02 | 0.02 | 0.01 | 0.02 | 0.01 | 0.02 | 0.01 | 0.02 | 0.01 | 0.01 | 0.01 | 0.01 |
| Mg | apfu | 0.02 | 0.02 | 0.01 | 0.02 | 0.03 | 0.02 | 0.02 | 0.01 | 0.01 | 0.01 | 0.01 | 0.00 | 0.07 | 0.07 | 0.06 | 0.08 |
| Ca | apfu | 3.01 | 3.02 | 3.01 | 3.07 | 2.97 | 3.05 | 3.01 | 3.04 | 3.00 | 2.97 | 3.01 | 2.97 | 2.96 | 2.95 | 2.98 | 2.97 |
| Ura | wt.% | 0.22 | 0.00 | 0.28 | 0.13 | 0.53 | 0.03 | 0.20 | 0.00 | 0.00 | 0.00 | 0.00 | 0.00 | 0.56 | 0.38 | 0.55 | 0.63 |
| Ad | wt.% | 44.12 | 33.82 | 37.23 | 41.57 | 52.89 | 48.92 | 40.09 | 37.34 | 37.24 | 44.90 | 39.03 | 30.54 | 26.49 | 22.28 | 22.22 | 21.89 |
| Pyr | wt.% | 0.69 | 0.52 | 0.43 | 0.51 | 0.90 | 0.63 | 0.81 | 0.49 | 0.26 | 0.37 | 0.44 | 0.13 | 2.27 | 2.27 | 2.10 | 2.59 |
| Spe | wt.% | 0.41 | 0.34 | 0.48 | 0.41 | 0.75 | 0.61 | 0.35 | 0.62 | 0.25 | 0.58 | 0.23 | 0.55 | 0.37 | 0.42 | 0.37 | 0.33 |
| Gro | wt.% | 54.55 | 65.32 | 61.57 | 57.37 | 44.93 | 49.81 | 58.56 | 61.55 | 62.24 | 54.14 | 60.30 | 68.79 | 70.32 | 74.66 | 74.76 | 74.56 |
| Alm | wt.% | 0.00 | 0.00 | 0.00 | 0.00 | 0.00 | 0.00 | 0.00 | 0.00 | 0.00 | 0.00 | 0.00 | 0.00 | 0.00 | 0.00 | 0.00 | 0.00 |

**Note**: Ura-Uvarovite; Ad-Andradite; Pyr-Pyrope; Spe-Spessartite; Gro-Grossular; Alm-Almandine.

**Table 2.** Electron microprobe analyses of pyroxene from the Da'anhe deposits.

| Sample | Unit | DAH-1-1 | DAH-1-2 | DAH-1-3 | DAH-1-4 | DAH-1-5 |
|---|---|---|---|---|---|---|
| $SiO_2$ | wt.% | 54.11 | 54.22 | 53.95 | 54.59 | 54.84 |
| $TiO_2$ | wt.% | 0.00 | 0.00 | 0.04 | 0.03 | 0.01 |
| $Al_2O_3$ | wt.% | 0.15 | 0.34 | 0.34 | 0.14 | 0.16 |
| $Cr_2O_3$ | wt.% | 0.03 | 0.05 | 0.08 | 0.03 | 0.02 |
| FeO | wt.% | 8.62 | 8.72 | 10.10 | 8.62 | 9.21 |
| MnO | wt.% | 0.33 | 0.13 | 0.31 | 0.33 | 0.35 |
| MgO | wt.% | 11.88 | 11.79 | 11.05 | 11.90 | 11.40 |
| CaO | wt.% | 25.51 | 25.24 | 24.99 | 25.28 | 24.98 |
| $Na_2O$ | wt.% | 0.11 | 0.18 | 0.16 | 0.08 | 0.10 |
| $K_2O$ | wt.% | 0.05 | 0.00 | 0.01 | 0.00 | 0.04 |
| Total | wt.% | 100.78 | 100.68 | 101.02 | 100.99 | 101.11 |
| **Based on 6 oxygen atoms.** | | | | | | |
| Al(iv) | apfu | 0.00 | 0.00 | 0.00 | 0.00 | 0.00 |
| Al(vi) | apfu | 0.01 | 0.01 | 0.01 | 0.01 | 0.01 |
| Ti | apfu | 0.00 | 0.00 | 0.00 | 0.00 | 0.00 |
| Cr | apfu | 0.00 | 0.00 | 0.00 | 0.00 | 0.00 |
| $Fe^{3+}$ | apfu | 0.00 | 0.00 | 0.00 | 0.00 | 0.00 |
| $Fe^{2+}$ | apfu | 0.27 | 0.27 | 0.32 | 0.27 | 0.29 |
| Mn | apfu | 0.01 | 0.00 | 0.01 | 0.01 | 0.01 |
| Mg | apfu | 0.66 | 0.65 | 0.61 | 0.66 | 0.63 |
| Ca | apfu | 1.02 | 1.01 | 1.00 | 1.00 | 0.99 |
| Na | apfu | 0.01 | 0.01 | 0.01 | 0.01 | 0.01 |
| K | apfu | 0.00 | 0.00 | 0.00 | 0.00 | 0.00 |
| Di | wt.% | 70.21 | 70.25 | 65.31 | 70.21 | 67.83 |
| Hd | wt.% | 28.67 | 29.30 | 33.65 | 28.70 | 30.99 |
| Jo | wt.% | 1.12 | 0.45 | 1.04 | 1.09 | 1.18 |
| Mn/Fe | —— | 0.02 | 0.01 | 0.02 | 0.02 | 0.02 |

**Note**: Di-Diopside; Hd-Hedenbergite; Jo-Johannsenite.

*5.2. Results of FI Analysis*

5.2.1. Petrographic and Microthermometric Results for FIs

In this study, a petrographic microscope was used to observe the FIs in garnet, diopside, epidote, quartz and calcite in the Da'anhe mining area. Among them, the FIs in quartz and calcite were the most common, followed by those in garnet and diopside, and the FIs in epidote were the least common. Most of the FIs were primary FIs, with a small number of secondary FIs developed along mineral fissures. The sizes of the FIs vary and are concentrated in the range of 3~15 μm, and the maximum size is more than 20 μm. The morphologies are mainly elliptical and irregular, with minor long strip and spindle morphologies. According to the phase characteristics of FIs at room temperature, the FIs in the Da'anhe gold deposit can be divided into five types: daughter mineral-bearing three-phase FIs (S type), liquid-rich FIs (W1 type), gas-rich FIs (W2 type), pure gas FIs (PV) and pure liquid FIs (PL).

(1)  S-type FIs: S-type FIs developed in the early skarn stage and late skarn stage and those that exhibit oval, spindle or irregular shapes. These FIs in garnet are larger (Figure 7A), and the FIs in epidote are smaller. The overall sizes range from 3~20 μm, and the gas-liquid ratios vary from 15% to 35%. Combined with the morphology of transparent daughter minerals in FIs and the findings of previous research [20,21], we believe that the crystalline minerals are mainly halite and sylvinite. These FIs are generally less common, accounting for approximately 10% of the total number of inclusions, and coexist with the W1-, W2- and PV-type inclusions (Figure 7A,C).

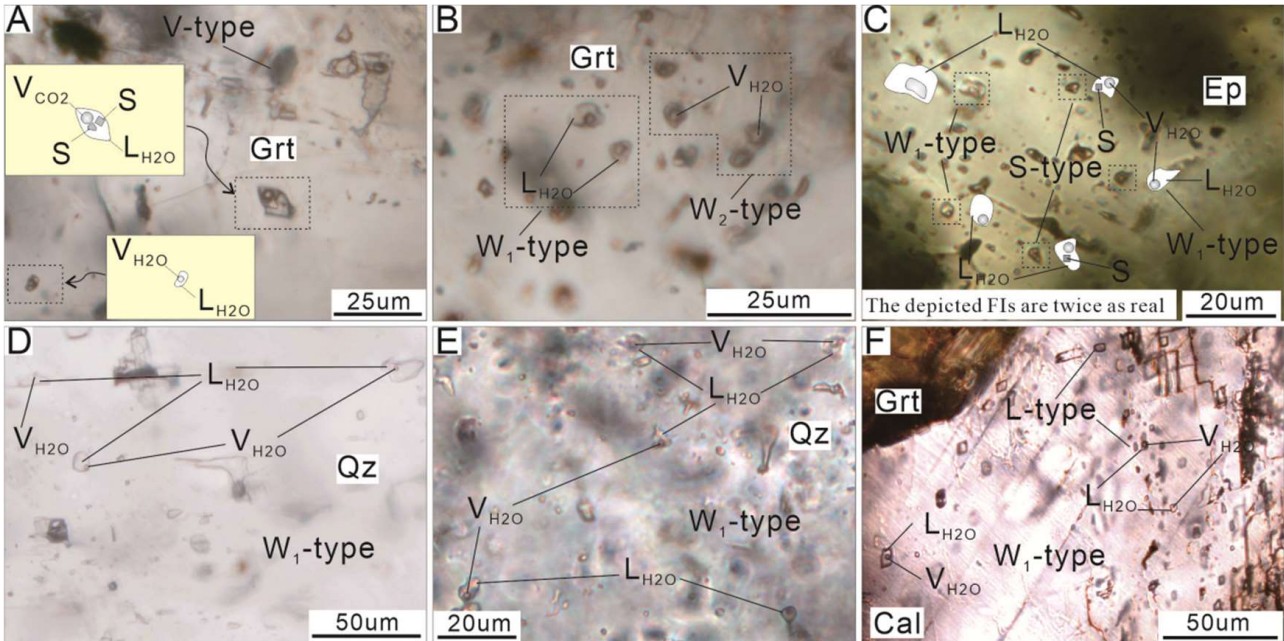

**Figure 7.** Microscopic photographs of fluid inclusions in the Da'anhe deposit. (**A**): W1-, S- and PV-type FIs coexist in garnet; (**B**): W1- and W2-type FIs coexist in garnet; (**C**): W1- and S-type FIs coexist in epidote; (**D**): W1-type FIs in early quartz-sulfide stage; (**E**): W1-type FIs in quartz of late quartz-sulfide stage; (**F**): W1- and PL-type FIs coexist in calcite of quartz-carbonate stage.

(2) W1-type FIs: These FIs formed throughout the metallogenic stage and are observed in garnet, diopside, epidote, quartz and calcite, especially in quartz. Most of the FIs are oval and irregularly shaped, range from 3~18 μm, and account for approximately 60% of the total number of FIs. The gas-liquid ratios of W1-type inclusions vary greatly from 5%~40%, and the gas-liquid ratios in the early quartz-sulfide stage inclusions are the lowest (Figure 7A–F).

(3) W2-type FIs: These FIs developed during all stages, except the quartz-carbonate stage, and appear in large quantities in quartz. The FIs are mostly uniform in size, with oval and irregular shapes and a small number of long strips, range in size from 5 to 15 μm, and account for approximately 25% of the total number of inclusions. The gas-liquid ratios of the W2 inclusions range between 65% and 80% (Figure 7B).

(4) PV-type and PL-type FIs: These two types of inclusions are less common in each mineralization stage and account for approximately 5% of the total number of inclusions. Among them, the PV-type FIs mainly developed during the skarn stage, while the PL-type FIs mainly developed during the quartz carbonate stage of late mineralization. The FIs are mostly elliptical and cuboid shaped, and a few are irregular or long strips, with sizes ranging from 2 to 15 μm (Figure 7A,F).

On the basis of petrographic observations, temperature measurements during the heating and cooling stages were carried out for the FIs in each mineralization. Most of the FIs used for these temperature measurement stages appear in groups, indicating that they are primary FIs. A total of 164 uniform temperature data points were obtained. The test results are listed in Table 3.

**Table 3.** Microscopic thermometry results of fluid inclusions in the Da'anhe gold deposit.

| Stage | Sample | Minerals | Type | Gas-Liquid Ratio ($V_G/V_T$%) | $T_d$ (°C) | $T_{m\text{-}ice}$ (°C) | $T_h$ (°C) | Salinity (wt.% NaCl Equiv.) | Density (g/cm³) | Number | Pressure (MPa) | Depth (Km) |
|---|---|---|---|---|---|---|---|---|---|---|---|---|
| $I_1$ | **DAH-1** | Grt | **W1** | 10~35 | | −12.7~−7.8 | 447~515 | 11.5~16.7 | 0.56~0.61 | 11 | 49.5~64.1 | 5.60~6.46 |
| | | | W2 | | | −10.8~−8.8 | 468~511 | 12.7~14.8 | 0.55~0.59 | 8 | 52.5~60.8 | 5.79~6.27 |
| | | | S | 15~30 | 393~431 | | 477~503 | 46.7~51.0 | 1.56~1.70 | 5 | | |
| | DAH-1 | Di | W1 | 15~40 | | −13.1~−8.5 | 420~520 | 12.3~17.1 | 0.57~0.66 | 11 | 46.7~65.3 | 5.43~6.52 |
| | | | W2 | | | −11.7~−7.6 | 453~511 | 11.2~15.8 | 0.57~0.58 | 6 | 48.7~62.2 | 5.56~6.35 |
| | | | S | 15~30 | 409~431 | | 481~505 | 38.0~42.0 | 1.62~1.70 | 4 | | |
| $I_2$ | DAH-2 | Ep | W1 | 10~35 | | −8.5~−5.8 | 358~431 | 8.94~12.3 | 0.64~0.72 | 17 | 35.6~47.9 | 3.56~5.51 |
| | | | W2 | | | −7.3~−5.7 | 356~418 | 8.81~10.9 | 0.63~0.72 | 12 | 35.2~43.6 | 3.52~5.24 |
| | | | S | 10~30 | 336~366 | | 385~428 | 41.1~43.9 | 1.45~1.50 | 5 | | |
| $II_1$ | DAH-3 | $Q_1$ | W1 | 15~35 | | −6.3~−3.7 | 279~366 | 5.99~9.60 | 0.71~0.80 | 19 | 24.5~37.3 | 2.45~3.73 |
| | | | W2 | 55~60 | | −6.4~−4.4 | 305~357 | 7.01~9.73 | 0.73~0.78 | 10 | 28.2~36.5 | 2.82~3.65 |
| $II_2$ | DAH-4 | $Q_2$ | W1 | 10~30 | | −5.7~−2.5 | 226~305 | 4.17~8.81 | 0.80~0.87 | 15 | 18.0~30.2 | 1.8~3.02 |
| | | | W2 | 60~70 | | −4.2~−2.5 | 247~300 | 4.17~5.62 | 0.78~0.84 | 11 | 20.0~27.2 | 2.0~2.72 |
| $II_3$ | DAH-5 | Cal | W1 | 15~25 | | −2.0~−0.5 | 137~247 | 0.87~3.37 | 0.82~0.94 | 30 | 9.25~18.8 | 0.93~1.88 |

**Note**: Grt-Garnet; Di-Diopside; Ep-Epidote; $Q_1$-Quartz from early quartz sulfide stage; $Q_2$-Quartz from late quartz sulfide stage; Cal-Calcite. $T_d$-disappearance temperature of daughter minerals; $T_{m\text{-}ice}$-freezing point temperatures; $T_h$-homogenization temperature; $V_G$-Volume of gas; $V_T$-Total Volume.

Early skarn stage: The FIs used for temperature measurements during the heating and cooling stages include garnet and diopside. The FIs are mainly W1- and S-type FIs, with a small number of W2- and PL-type FIs. In the same field of view, W1-, S- and PV-type FIs coexist (Figure 7A), indicating that the fluid underwent boiling during the evolution process. When the W1-type FIs are heated to 420~520 °C, they homogenize to the liquid phase (homogenization temperature, $T_h$). The freezing point temperatures ($T_{m\text{-}ice}$) of this type of inclusion vary from −13.1 °C to −7.8 °C and are obtained by cooling. After calculation, the corresponding salinities range from 11.5~17.1 wt.% NaCl equiv., and the densities range from 0.56~0.66 g/cm³. When the W2-type FIs are heated to 453~511 °C, they uniformly transform to the gas phase. The $T_{m\text{-}ice}$ values of the W2-type FIs vary from −11.7 to −7.6 °C and are obtained by cooling. The salinities of the corresponding fluids range from 11.2~15.8 wt.% NaCl equiv. and the densities range from 0.55~0.59 g/cm³. The daughter minerals in the S-type FIs disappear when heated to 393~431 °C (disappearance temperature of daughter minerals, $T_d$) and homogenize to the liquid phase when heated to 477~503 °C. Based on these results, the salinities and densities of the fluids are 38.0~51.0 wt.% NaCl equiv. and 1.56~1.70 g/cm³, respectively (Figures 7 and 8A–D).

Late skarn stage: The FIs used for temperature measurements using the heating and cooling stage are from epidote. The FIs that developed in this stage were mainly W1, W2 and S-type FIs. The coexistence of different types of FIs in the same field of view is observed in epidote (Figure 7C), indicating that fluid boiling also occurred during this stage. When the W1-type FIs homogenize into the liquid phase, the temperature is 358~431 °C, the freezing point temperatures vary from −8.5 to −5.8 °C, the corresponding salinities are 8.94~12.3 wt.% NaCl equiv., and the densities are 0.64~0.72 g/cm³ (Figures 7 and 8E,F). W2-type FIs homogenize into the gas phase at temperatures of 356~418 °C, the freezing point temperature varies from −7.3 to −5.7 °C, and their corresponding salinities are 8.81~10.9 wt.% NaCl equiv. and the density is 0.63~0.72 g/cm³. The daughter minerals in the S-type inclusions disappear at 336~366 °C, and they homogenize into the liquid phase at 385 to 428 °C. After calculation, their corresponding salinities range from 41.1~43.9 wt.% NaCl equiv., and their densities are 1.45~1.50 g/cm³ (Figures 7 and 8E,F).

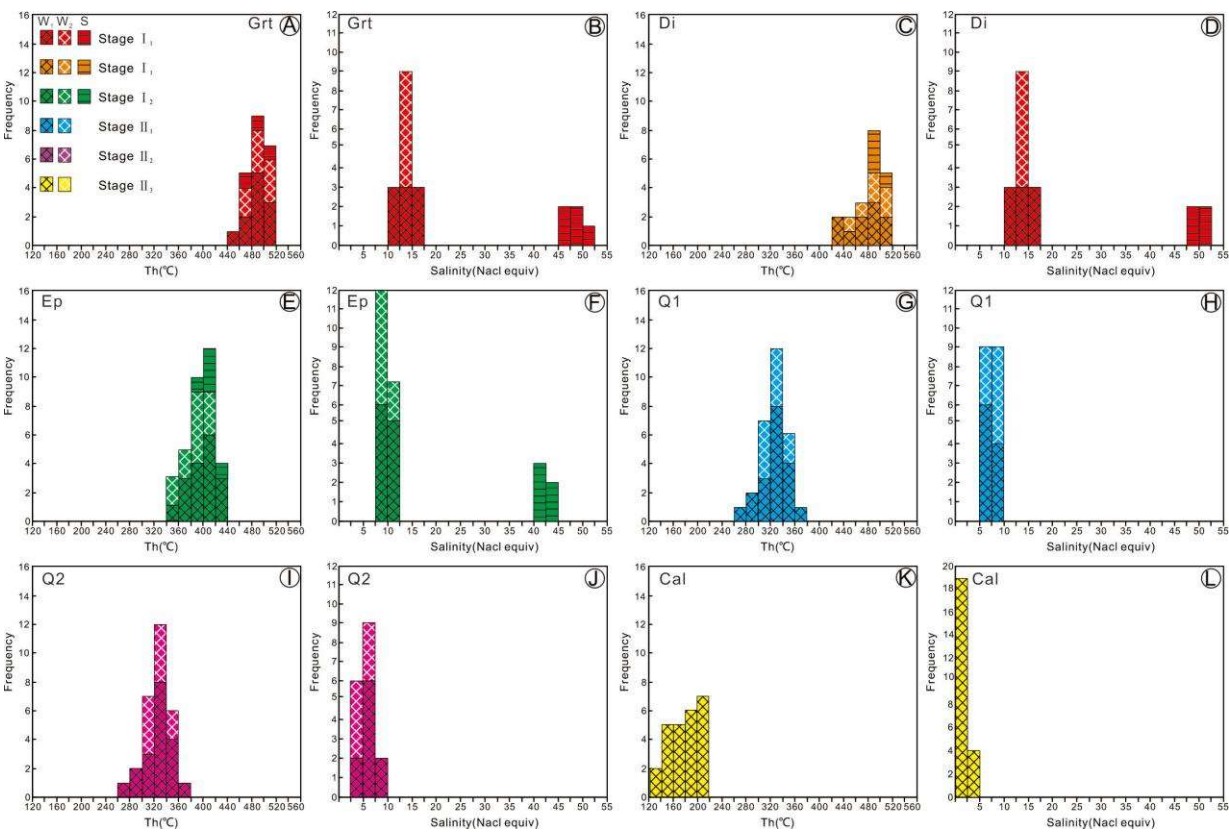

**Figure 8.** Histograms of total homogenization temperatures ($T_h$) and salinities of FIs in different stages. (**A–D**): Stage $I_1$; (**E,F**): Stage $I_2$; (**G,H**): Stage $II_1$; (**I,J**): Stage $II_2$; (**K,L**): Stage $II_3$.

Early quartz-sulfide stage: The FIs used for temperature measurements using the heating and cooling stage are from quartz. The FIs are mainly W1 and W2 type FIs, and no S-type FIs are found (Figure 7). The W1-type FIs were heated and homogenized to the liquid phase. The homogenization temperatures range from 279 to 366 °C, the freezing point temperatures vary from −6.3 to −3.7 °C, the calculated corresponding salinities are 5.99~9.60 wt.% NaCl equiv., and the densities are 0.71~0.80 g/cm³. For the W2-type FIs, the homogenization temperatures, freezing point temperatures, and corresponding salinities and densities are 305~357 °C, −6.4~−4.4 °C, 7.01~9.73 wt.% NaCl equiv., and 0.73~0.78 g/cm³, respectively (Figure 8G,H).

Late quartz-sulfide stage: The FIs used for temperature measurements using the heating and cooling stage are from quartz. Petrography revealed that the types of FIs in this stage are similar to those in the early quartz-sulfide stage, i.e., mainly W1- and W2-type FIs (Figure 7E). In this stage, the W1-type FIs were heated and homogenized to the liquid phase. The homogenization temperatures range from 226 to 305 °C, and the freezing point temperatures vary from −5.7 to −2.5 °C. After calculation, their corresponding salinities range from 4.17~8.81 wt.% NaCl equiv., and their densities range from 0.80~0.87 g/cm³. For the W2-type FIs, the homogenization temperatures range from 247~300 °C, the freezing point temperatures vary from −4.2 to −2.5 °C, their corresponding salinities are 4.17~5.62 wt.% NaCl equiv., and their densities are 0.78~0.84 g/cm³ (Figure 8I,J).

Quartz-carbonate stage: The FIs developed in calcite veins were selected for microscopic analysis. The petrography shows that only W1- and PL-type FIs developed during this stage. The homogenization temperatures of the W1-type FIs range from 137 to 247 °C, and the freezing point temperatures vary from −2.0~−0.5 °C. After calculation, their corresponding salinities are 0.87~3.37 wt.% NaCl equiv., and their fluid densities are 0.82~0.94 g/cm³ (Figure 8K,L).

### 5.2.2. Laser Raman Spectral Analysis Results

To further understand the compositions of the FIs, the authors of this paper selected representative FIs from different mineralization stages of the Da'anhe gold deposit for Laser Raman spectral analysis. Because W2-type FIs are not well developed, W1-type FIs were used.

The Laser Raman spectral results show that the gas phase components of the W1-type FIs in the early skarn stage are $CH_4$ and $H_2O$ (Figure 9A,B). The $CH_4$ and $H_2O$ peaks of the W1-type FIs in the late skarn stage are weaker than those in the early skarn stage, and $CO_2$ is present (Figure 9C). The gas phase composition of the W1-type FIs in the early quartz-sulfide stage is mainly $H_2O$ with a small amount of $CH_4$ (Figure 9D). In the late quartz-sulfide stage and quartz-carbonate stage, the gas phase components in the W1-type FIs are $CH_4$ and $H_2O$, and the peak intensity of $CH_4$ gradually increases (Figure 9E,F).

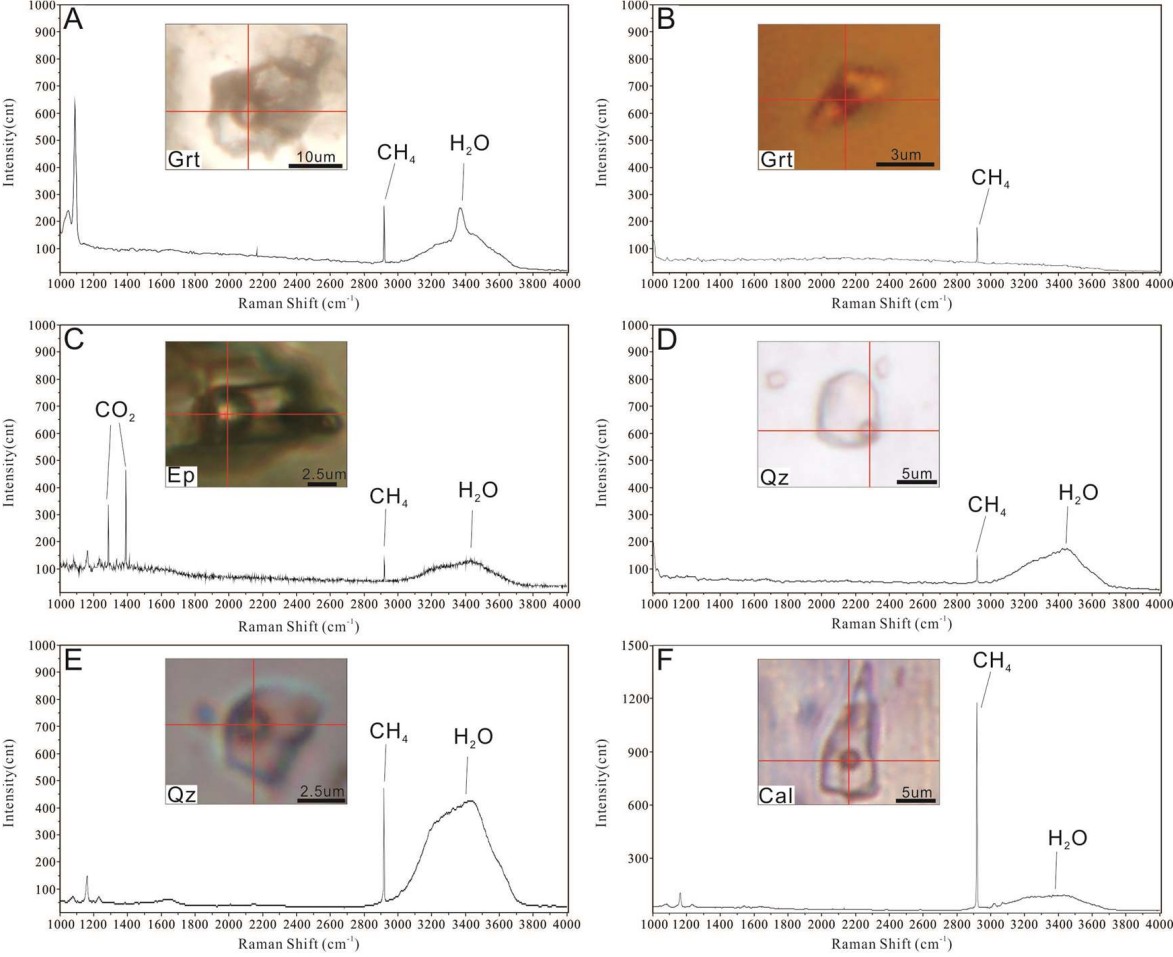

**Figure 9.** Representative Laser Raman spectra of FIs in garnet, epidote, quartz and calcite grains of different stages from the Da'anhe deposit. (**A**,**B**): Spectrum for vapor phase of W1−type FIs hosted in garnet grain of the dry skarn stage; (**C**): Spectrum for vapor phase of W1−type FIs hosted in epidote grain of the wet skarn stage; (**D**): Spectrum for vapor phase of W1−type FIs hosted in quartz grain of the early quartz sulfide stage; (**E**): Spectrum for vapor phase of W1−type FIs hosted in quartz grain of the late quartz sulfide stage; (**F**): Spectrum for vapor phase of W1−type FIs hosted in calcite grain of the quartz-carbonate stage.

In general, the gas components in the ore-forming fluid of the Da'anhe deposit are mainly $CH_4$, $H_2O$ and a small amount of $CO_2$. The process of mineralization is commonly accompanied by a transition from an oxidizing environment to a reducing environment. Therefore, we suggest that the large amount of $CH_4$ present during the skarn stage was

provided by carbonate rocks, and that the $CO_2$ in the fluid may have escaped in large quantities. As the mineralization entered the early quartz sulfide stage, the abundance of $CH_4$ gradually decreased, and that of $CO_2$ gradually increased. During the late quartz sulfide stage and quartz carbonate stage, a large amount of $CO_2$ in the fluid was consumed, the ore-forming system transformed into a reducing environment, and large amounts of $CH_4$ accumulated. Therefore, the initial ore-forming fluid was a $H_2O$-$CO_2$-NaCl system (Figure 9).

*5.3. H-O Isotope Analytical Results*

The results of mineral H-O isotope tests for different metallogenic stages in the Da'anhe gold deposit (Figure 10, Table 4) [2,43] reveal that the $\delta D_{H2O}$ values of garnet samples in the early skarn stage vary from −89.1‰ to −85.9‰, and the $\delta^{18}O_{V\text{-}SMOW}$ values vary from 6.1‰ to 7.0‰. Combined with the results of FI temperature measurements (the average value is 490 °C), the $\delta^{18}O_{H2O}$ values of the fluid are calculated and vary from 7.57‰ to 8.47‰. In the late skarn stage, the $\delta D_{H2O}$ value of epidote is −73.3‰, and the $\delta^{18}O_{V\text{-}SMOW}$ value is 5.4‰. After calculation, the $\delta^{18}O_{H2O}$ value of the fluid is 5.71‰ ($T_h$ is 393 °C). The $\delta D_{H2O}$ values of the quartz samples in the late quartz-sulfide stage vary from −79.3‰ to −102.5‰, and the $\delta^{18}O_{V\text{-}SMOW}$ values range from 8.7‰ to 14.1‰. After recalculation, the $\delta^{18}O_{H2O}$ values of the fluid vary from 0.25‰ to 5.65‰ ($T_h$ is 261 °C). The $\delta D_{H2O}$ values of calcite samples in the quartz-carbonate stage are concentrated at −116‰, and the $\delta^{18}O_{V\text{-}SMOW}$ value is 11.6‰. The calculated $\delta^{18}O_{H2O}$ value of the fluid is −0.77‰ ($T_h$ is 190 °C).

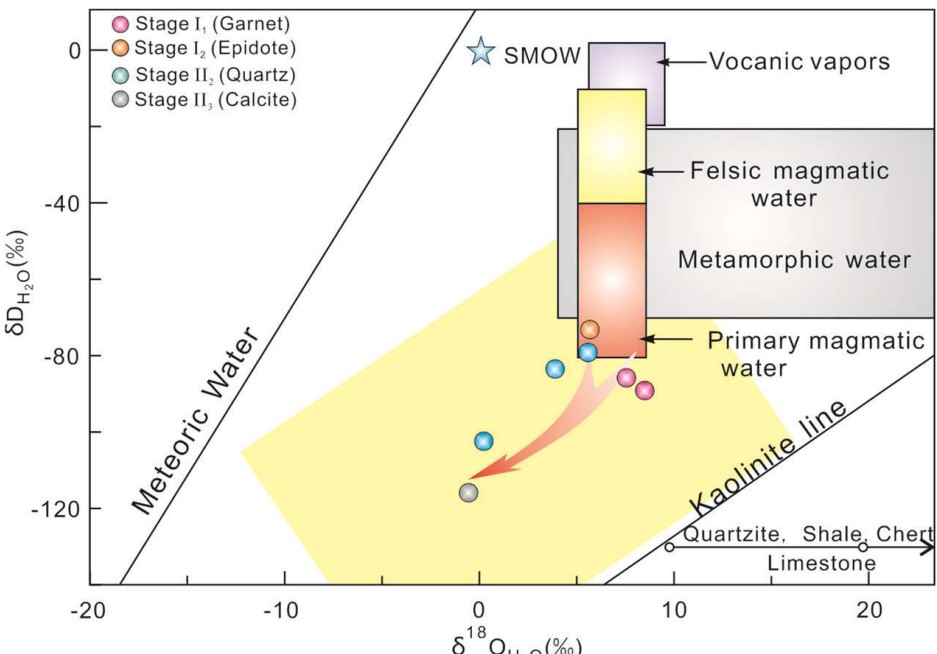

**Figure 10.** Hydrogen and oxygen isotopic compositions of the ore−forming fluid in the Da'anhe deposit (data taken from Table 4, modified after Hedenquist and Lowenstern [2]).

**Table 4.** Hydrogen and oxygen isotopic compositions of fluids from the Da'anhe Au deposit.

| Sample | Mineral | $Th$ (°C) | $\delta^{18}O_{V\text{-SMOW}}$ (‰) | $\delta^{18}O_{H2O}$ (‰) | $\delta D_{H2O}$ (‰) | Data Sources |
|---|---|---|---|---|---|---|
| 19DAH-2 | Garnet | 490 | 6.1 | 7.57 | −85.9 | This study |
| 19DAH-3 | | | 7.0 | 8.47 | −89.1 | |
| 19DAH-3 | Epidote | 393 | 5.4 | 5.71 | −73.3 | |
| 19DAH-7 | Quartz | 261 | 8.7 | 0.25 | −102.5 | |
| DKDA3 | | | 12.4 | 3.95 | −83.4 | [43] |
| DKDA4 | | | 14.1 | 5.65 | −79.3 | |
| 19DAH-7 | Calcite | 190 | 11.6 | −0.77 | −116 | This study |

## 6. Discussion

### 6.1. Implications of Skarn Minerals for Mineralization

The chemical composition of skarn minerals can be characterized by physical and chemical conditions, such as temperature, pressure, depth, pH and oxygen fugacity during their formation, which can provide important information on the formation environment of deposits [44–46]. It is generally believed that andradite and magnesium-rich diopside are formed in high-temperature environments with strong oxidation reactions and weak acidity, while grossular and hedenbergite are often formed in environments with high temperature, weak oxidation, reduction and alkalinity [45,47,48]. The garnet in the Da'anhe gold deposit is mainly grossular (Figure 6C), and pyroxene belongs to the diopside-hedenbergite solid solution series (Figure 6D). Both coexist in the outer belt of the skarn, and the pyroxene content is greater than the garnet content. The above results show that the Da'anhe deposit formed in a reducing environment related to an intrusive body and is a typical calcareous reduced skarn Au deposit. In addition, our team conducted electron probe analysis on the hornblende and biotite in the gabbroic diorite, and the results showed that the gabbro-diorite magma related to mineralization had reducing characteristics ($\Delta$FMQ = −0.95~0.60, with an average value of 0.17; log($fO_2$) = −15.65~−14.68, with an average value of −15.11 [49]), which indicates that the Da'anhe deposit is a reduced skarn gold deposit.

### 6.2. Source of the Ore-Forming Materials

Previous studies have shown that skarn deposits in the Lesser Xing'an Range generally lack sulfate minerals [49]. Therefore, the $\delta^{34}S$ values of the metal sulfide minerals in the ore can be used to represent the $\delta^{34}S$ values of the ore-forming fluid of the deposit [50]. To determine the source of ore-forming materials in the Da'anhe gold deposit, S-Pb isotope data from metal sulfides in the Da'anhe gold deposit were collected and are listed in Tables 5 and 6. The S-Pb isotope data show that the $\delta^{34}S$ values of pyrrhotite in the Da'anhe deposit vary from −0.6‰ to 0.5‰ (with an average of 0.10‰), the $\delta^{34}S$ values of pyrite vary from −2.9‰ to −0.6‰ (with an average of −3.07‰), and the arithmetic average of the sulfides is −0.83‰. The $\delta^{34}S$ values of sulfides in the Da'anhe deposit are within the range of typical porphyry-skarn deposits (−5.0‰ to 5.0‰, [51]) (Figure 11) [50,52], indicating that the sulfur in the Da'anhe deposit may have mainly come from the magmatic system. The Pb isotope values of the metal sulfides in the Da'anhe deposit vary greatly, and the $^{206}Pb/^{204}Pb$, $^{207}Pb/^{204}Pb$ and $^{208}Pb/^{204}Pb$ ratios are 18.099~18.568, 15.532~15.580 and 37.909~38.176, respectively. In the $^{207}Pb/^{204}Pb$-$^{206}Pb/^{204}Pb$ projection (Figure 12) [53], the sample points are mainly located in the lower crust area. According to the $^{208}Pb/^{204}Pb$-$^{206}Pb/^{204}Pb$ diagram, the sample points are located mainly on both sides of the orogenic belt evolution line, and some of them are located on the upper crust evolution line, which indicates that the Pb in the Da'anhe sulfide has the characteristics of mixed lead. These results imply that the ore-forming materials may have been derived from crust-mantle mixing. The Sr-Nd isotopes of the medium-grained gabbroic diorite related to the mineral-

ization of the Da'anhe gold deposit have been studied by Yang et al. [54] ($\varepsilon$Nd(t) = 0.8~1.9, $\varepsilon$Sr(t) = 75.9~77.2, and ($^{87}$Sr/$^{86}$Sr)$_i$ = 0.70962~0.70972, $T_{DM2}$ = 815–1076 Ma); their results indicate that the mineralization originated from Neoproterozoic lower crust metasomatized by fluids or weakly enriched mantle and that the ascending material was contaminated by young crustal materials during the process of ascent and emplacement [54]. This result further shows that the ore-forming material of the Da'anhe deposit was derived from gabbro-dioritic magma.

**Table 5.** Sulfur isotopic composition of sulfides from the Da'anhe Au deposit.

| Sample | Mineral | $\delta^{34}$S‰ | Data Sources |
|---|---|---|---|
| Dah4-Pyr | Pyr | 0.5 | |
| Dah5-Pyr | Pyr | 0.5 | |
| Dah7 | Pyr | −0.6 | |
| Dah4-Py | Py | −0.6 | [21] |
| Dah5-Py | Py | −2.9 | |
| Dah6-Py | Py | −2.7 | |
| Dah6-Pyr | Pyr | 0 | |

**Note**: Pyr-Pyrrhotite; Py-Pyrite.

**Table 6.** Lead isotopic composition of sulfides from the Da'anhe Au deposit.

| Sample | Mineral | $^{206}$Pb/$^{204}$Pb | $^{207}$Pb/$^{204}$Pb | $^{208}$Pb/$^{204}$Pb | Data Sources |
|---|---|---|---|---|---|
| Dah4-Pyr | Pyr | 18.308 | 15.543 | 38.139 | |
| Dah4-Py | Py | 18.508 | 15.578 | 38.176 | |
| Dah5-Pyr | Pyr | 18.303 | 15.536 | 38.125 | |
| Dah5-Py | Py | 18.099 | 15.556 | 38.071 | [21] |
| Dah6Py | Py | 18.317 | 15.532 | 38.098 | |
| Dah6Pyr | Pyr | 18.568 | 15.580 | 37.909 | |
| Dah7 | Pyr | 18.455 | 15.577 | 38.103 | |

**Note**: Pyr-Pyrrhotite; Py-Pyrite.

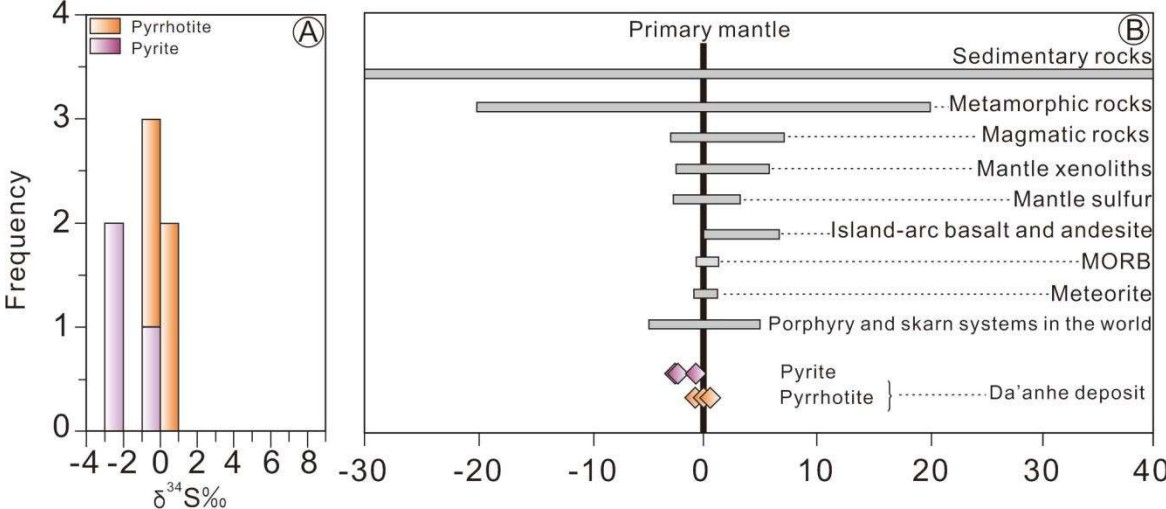

**Figure 11.** (**A**) Histogram of the sulfur isotopic compositions of sulfides from the Da'anhe deposit; (**B**) Ranges of $\delta^{34}$S values of selected geologically important sulfur reservoirs [50,52] and the Da'anhe deposit (data taken from Table 5). The orange and purple squares represent magnetite and pyrite, respectively. The gray squares represent different geologically important sulfur reservoirs.

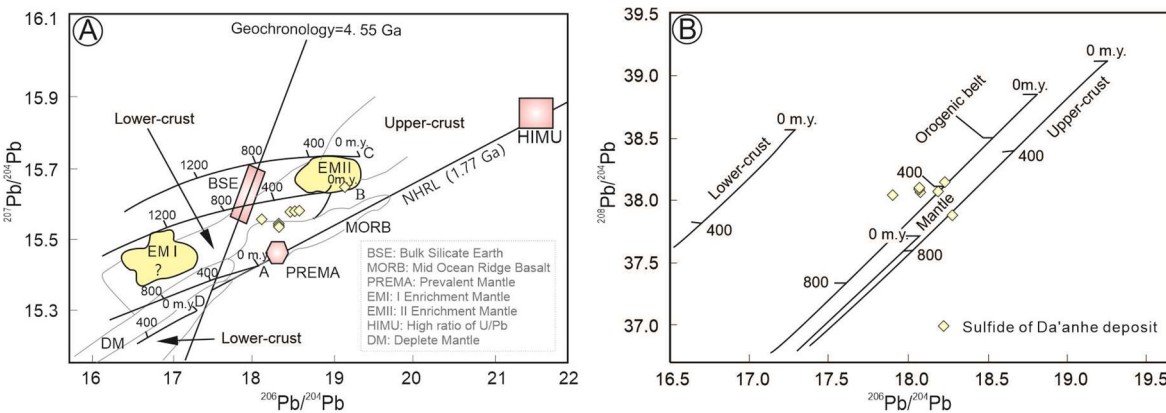

**Figure 12.** Diagram of lead isotopic compositions of the Da'anhe deposit. (**A**)—$^{207}$Pb/$^{204}$Pb v.s. $^{206}$Pb/$^{204}$Pb; (**B**)—$^{208}$Pb/$^{204}$Pb v.s. $^{206}$Pb/$^{204}$Pb (data taken from Table 6, modified after Zartman and Doe [53]).

### 6.3. Source and Evolution of Ore-Forming Fluids

For the source of ore-forming fluids in skarn deposits, magmatic water is generally the dominant source, followed by meteoric water [55,56], and the duration and amount of meteoric water can vary according to the geological characteristics of different deposits. Several geologists [55,57] have suggested that the different alteration characteristics of the early skarn stage and late skarn stage reflect the dominant roles of magmatic water and meteoric water, respectively. Meinert et al. [58,59] suggested that these two processes may be the result of the evolution of a single magmatic hydrothermal system. The FIs in the skarn minerals of the Da'anhe gold deposit are mainly W1- and W2-type FIs and a small number of S-type FIs. According to the crystal form, the daughter crystals are mainly halite and sylvinite. Laser Raman spectroscopy shows that the gas phases involved in the early skarn stage were $H_2O$ and $CH_4$ and that $CO_2$-containing FIs began to appear in the late skarn stage. The homogenization temperatures of the FIs that developed in the early skarn and late skarn stages are 393~515 °C and 336~431 °C, respectively, and their corresponding salinities are 11.2~51.0 wt.% NaCl equiv. and 8.81~43.9 wt.% NaCl equiv., respectively. The above characteristics indicate that the initial ore-forming fluid was an $H_2O$-NaCl-$CO_2$ system with a high temperature and high salinity. The main types of FIs in the quartz-sulfide period are W1 and W2. Laser Raman analysis revealed that the gas phases in the FIs are mainly $H_2O$ and $CH_4$. The homogenization temperatures of the FIs from the early to late stages are 279~366 °C, 226~305 °C and 137~247 °C. The corresponding salinities are 5.99~9.73 wt.% NaCl equiv., 4.17~8.81 wt.% NaCl equiv. and 0.87~3.37 wt.% NaCl equiv., respectively. The above characteristics indicate that the ore-forming fluids in this period were part of a medium- to low temperature and low salinity $H_2O$-$CH_4$ system. The H-O isotope data show that the $\delta D_{H2O}$ and $\delta^{18}O_{H2O}$ values ($\delta D_{H2O} = -89.1‰ \sim -73.3‰$; $\delta^{18}O_{H2O} = 5.71‰ \sim 8.47‰$) of the skarn minerals are close to the range of magmatic water during the skarn subperiod (Figure 10) [2], and the $\delta D_{H2O}$ and $\delta^{18}O_{H2O}$ values ($\delta D_{H2O} = -79.3‰ \sim -116‰$; $\delta^{18}O_{H2O} = -0.77‰ \sim 5.65‰$) of quartz in the quartz-sulfide period show a trend of gradual migration towards those of meteoric water (Figure 10) [2]. These results indicate that the initial ore-forming fluid of the Da'anhe gold deposit was derived from magmatic water that dissolved into the gabbro-dioritic magma. With continuous mineralization, the fluid mixed with meteoric water during the early stage of the quartz-sulfide period, and the mixing proportion gradually increased during the subsequent mineralization process until the end of mineralization at which point the system was dominated by meteoric water.

### 6.4. Metallogenic Pressure and Depth Estimation

To better understand the physical and chemical conditions involved in the mineralization process of the Da'anhe deposit, we calculated the pressure and depth at each stage.

Previous studies [60,61] have suggested that the trapping pressure can be estimated only when the exact trapping temperature is known or when FIs are trapped during phase separation. The coexistence of S-type, W1-type and W2-type FIs with similar temperature ranges occurred during the skarn period of the Da'anhe gold deposit, indicating that fluid immiscibility occurred during the early stage of mineralization. Previous studies have shown that cooling to a temperature below 400 °C leads to a transition from ductile to brittle rock behaviour, and therefore causes a change from lithostatic to hydrostatic pressure conditions [62]. The FIs in the quartz-sulfide period formed below 400 °C (137–366 °C), which indicates that the metallogenic system changed from lithostatic to hydrostatic pressure conditions. Therefore, the trapping pressures and depth were estimated using Flincor software [60]. The calculated results show that the metallogenic pressures during each mineralization stage ($I_1$, $I_2$, $II_1$, $II_2$ and $II_3$) in the Da'anhe deposit were 46.7~65.3 MPa, 35.2~47.9 MPa, 24.5~37.3 MPa, 18.0~30.2 MPa and 9.25~18.8 MPa, corresponding to mineralization depths of 5.43~6.52 km (average of 6.03 km), 3.52~5.51 km (average of 4.44 km), 2.45~3.73 km (average of 3.11 km), 1.80~3.02 km (average of 2.27 km) and 0.93~1.88 km (average of 1.36 km), respectively. As mentioned above, gold precipitated mainly during the quartz sulfide stage, so the depth of gold deposit formation ranged from 2.27~3.11 km. At present, the orebodies are close to the surface (<500 m), indicating strong uplift and denudation after the mineralization period. This result also corresponds to the denudation depth of approximately 3.0 km in the Lesser Xing'an Range since the Jurassic [6]. Based on the above characteristics, we suggest that the Da'anhe deposit has undergone severe erosion, making it difficult to identify industrial ore bodies at depth (Figure 13).

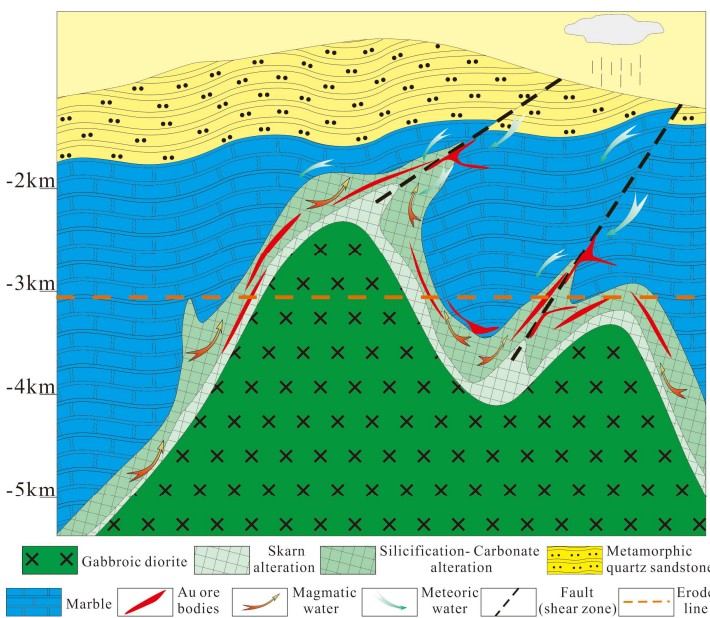

**Figure 13.** Simplified sketch that metallogenic model of the Da'anhe Au deposit.

### 6.5. Metallogenic Mechanism and Process

In magmatic-hydrothermal metallogenic systems, fluid boiling and fluid mixing are considered the most effective ways to restrict metal enrichment and precipitation [63]. The S-Pb isotopic characteristics of the metal sulfides and the H-O isotopic characteristics of the gangue minerals indicate that the initial ore-forming materials and ore-forming fluids of the Da'anhe gold deposit originated from gabbro-dioritic magma related to mineralization (Figures 10–12). The temperature of the magma chamber in the deep part of the Da'anhe deposit was very high, and the surrounding rock exhibited ductile

characteristics, thereby preventing interactions between the magma solution fluid and the external fluid. [62]. When the initial ore-bearing fluid rose to the shallow part of the upper crust along with the magma and met the carbonate (approximately 6.03 km), ore-bearing fluids with high temperatures ($T = 393{\sim}520\ ^{\circ}C$) and intermediate to high salinities ($S = 11.2\%{\sim}51.0\%$ NaCl equiv.) coexisted, and the reducing fluid underwent double metasomatism with the carbonate to form grossular and diopside (Stage $I_1$) (Figure 4C,D). The chemical reactions are as follows:

$$3CaCO_3 + Al_2O_3 + 3SiO_2 = Ca_3Al_2(SiO_4)_3(\text{grossular}) + 3CO_2\uparrow$$

$$CaCO_3 + MgCO_3 + 2SiO_2 = CaMgSi_2O_6(\text{diopside}) + 2CO_2\uparrow$$

Garnet and pyroxene contain three types of FIs (Figure 7A,B): S-type, W1-type and W2-type FIs, indicating that immiscibility caused by fluid boiling occurred in the early stage of mineralization. This process resulted in phase separation and the coexistence of different types of FIs in similar temperature ranges (Figure 8A–D) [64–66]. In the late skarn stage (Stage $I_2$), the ore-bearing hydrothermal solution continued to rise (reaching approximately 4.44 km). Although the overall temperature of the hydrothermal solution decreased during this time, it was still part of the hydrothermal system with high temperatures ($T = 336{\sim}431\ ^{\circ}C$) and high salinities ($S = 2.06\%{\sim}43.9\%$ NaCl equiv.) (Figure 8E,F). The FI evidence indicates that the ore-forming fluid began to shift to a subcritical state during this stage, and anhydrous skarn minerals, such as garnet and pyroxene, which formed during the early stage, underwent degenerative alteration and were replaced by water-bearing skarn minerals, such as epidote (Figure 5B,E), tremolite (Figures 4F and 5C), actinolite (Figure 4E) and chlorite. This process was accompanied by the precipitation of a small amount of magnetite (Figure 5G). The equations for the reactions in the late skarn stage is as follows [67]:

$$Ca_3Fe_2(SiO_4)_3\ (\text{andradite}) + 4CO_2 + 3/2O_2 = 4CaCO_3 + 3SiO_2 + Ca_2FeSi_3O_{13}\ (\text{epidote}) + Fe_3O_4\ (\text{magnetite})$$

$$5CaFe_2(SiO_4)_3\ (\text{hedenbergite}) + 3CO_2 + H_2O = Ca_2(Mg, Fe)_5Si_8O_{22}(OH)_2\ (\text{actinolite}) + 3CaCO_3 + 2SiO_2$$

Studies have shown that the process of skarn formation produces a large amount of free space, resulting in the release of pressure in the metallogenic system as well as boiling [10]. We found that the skarn minerals that formed early in the mining area were broken into breccia and cemented by quartz, chalcopyrite and pyrite (Figure 4G), indicating a sudden release of pressure during the skarn period. The petrographic study of FIs in the Da'anhe deposit shows that different types of FIs in skarn minerals (garnet, pyroxene and epidote) developed during the skarn stage (Figure 7A–C). According to the Th-salinity correlation diagram of FIs, high- and low-salinity FIs coexist within similar temperature intervals (Figure 8A–F), indicating that the ore-forming fluids experienced boiling [61].

The quartz-sulfide stage (Stage $II_1$ and Stage $II_2$) was the gold mineralization stage (Figures 4I and 5L). When the gold started to precipitate, the ore-forming fluid was part of a medium- to high-temperature and medium- to low-salinity reducing system ($T = 226{\sim}366\ ^{\circ}C$; $S = 4.17\%{\sim}9.73\%$ NaCl equiv., with $CH_4$ and $H_2O$ as the main gases) (Figures 8G–J and 9D,E). The ranges of temperature and pressure were relatively large. Studies have shown that gold migrates mainly in the form of chloride complexes in low-to high-temperature hydrothermal systems [68,69], and contributes to the control of the stability of metal complexes in ore-forming fluids because a decrease in temperature leads to the decomposition of metal chlorides and the precipitation of minerals [61,70]. Therefore, the solubility of Au in an ore-bearing fluid gradually decreases as the ore-forming fluid temperature decreases, and Au gradually precipitates in the ore-forming system.

In addition, fluid boiling during the skarn period caused the $H_2O$, $H_2S$, HCl and $CO_2$ components in the gas phase from escape to the original uniform fluid phase; this process resulted in an increase in the pH of the fluid, an increase in the concentrations of reduced sulfur and metal elements, and a decrease in the chloride ion concentration [71–73]. In

addition, the fluid that separated from the magma underwent a water-rock reaction with the surrounding rock (Figure 10), causing $CH_4$ to enter the mineralization system, leading to a continued decrease in the oxygen fugacity of the fluid [74–76]. Changes in the physical and chemical conditions of ore-forming fluids can promote the decomposition of metal complexes and eventually lead to the reaction of $Fe^{3+}$, $Cu^{2+}$, $Pb^{2+}$, $Zn^{2+}$ and other metal ions with $HS^-$ ions in solution, forming small amounts of metal sulfides (pyrite, pyrrhotite, chalcopyrite, sphalerite, etc.) [69,77].

In the quartz-carbonate stage (Stage $II_3$), after boiling, mixing and water-rock reactions in the previous stages, the initial ore-bearing hydrothermal fluid changed to a low-temperature and low-salinity system ($T$ = 137~247 °C; $S$ = 0.87%~3.37% NaCl equiv.) (Figure 8K,L). At this time, meteoric water entered and dominated the metallogenic system (Figure 10), and large amounts of quartz and calcite began to precipitate from the fluid, forming quartz-carbonate veins and cutting through skarns and orebodies (Figure 4L) and marking the end of mineralization.

Overall, the mineralization in the Da'anhe deposit experienced an evolutionary process from high temperatures and intermediate to high salinities to low temperatures and low salinities. Fluid boiling and mixing were the mechanisms driving gold precipitation.

### 6.6. Significant Indicators for Identifying Different Types of Skarn Deposits

Numerous skarn deposits have developed in NE China, and skarn gold deposits are mainly associated with copper deposits (such as the Laozuoshan Cu-Au deposit and Yuejinshan Cu-Au deposit). The Da'anhe deposit is the only independent skarn gold deposit in this region. By analysing the genesis and mineralization characteristics of the skarn deposits in the region, we suggest that the reasons for the formation of the Da'anhe independent skarn gold deposit are as follows.

Skarn deposits in northeastern China mainly formed in the late Permian-Early Triassic period (Baoshan Cu deposit [10]; Laozuoshan Cu-Au deposit [78]), Jurassic (Cuihongshan Fe-polymetallic deposit [14]; Ergu Fe-Zn polymetallic deposit [12]), and Early Cretaceous (Huanggangliang Fe-Sn deposit [79]; Haobugao Pb-Zn deposit [48]; Xieertala Fe-Zn deposit [80]; Tongshan Cu-Mo deposit [81]), while independent skarn gold deposits formed only in the Early Jurassic period (184–203 Ma, [19,20,43]. From the perspective of magmatic properties related to mineralization, the gabbro-dioritic magma related to the mineralization of the Da'anhe deposit originated from lower crust or mantle wedge metasomatized by subduction fluid, and such magmas are characterized by reducing conditions and a high water content ($\Delta$FMQ = 0.17, log($fO_2$) = −15.11, water content = 6.80 wt.%, [49]). The magmas related to the mineralization of the skarn Fe-Cu-Pb-Zn-Ag-W-Sn deposit were mostly felsic magmas (Chaobuleng Fe-polymetallic deposit [82]; Xulaojiugou Pb-Zn deposit [16]) or highly differentiated alkaline granitic magmas (Cuihongshan Fe-polymetallic deposit [14]), which were mainly derived from partial melting of the crust. These magmas were typically weakly oxidized to oxidized and had a high water content (Xieertala Fe-Zn deposit [80]; Ergu Fe-Cu polymetallic deposit [49]). In terms of the temperature during metal precipitation, the temperature during the precipitation of skarn Fe-Cu deposits is relatively high, corresponding to medium- to high-temperature mineralization (235–402 °C, Ergu Fe-Cu polymetallic deposit [11]; 200–350 °C, Handagai Fe-Cu deposit [83]). The temperature during the precipitation of skarn Pb-Zn-Ag deposits varies significantly (Xiaoxilin Pb-Zn deposit, 188.6–398.3 °C [43]; Baiyinnuoer, ~350 °C [84]), The temperature at which gold precipitated in the Da'anhe deposit was mainly in the medium temperature range (226–305 °C). Therefore, we propose that the properties of the magma and ore-forming fluids may contribute to the specific mineralization of skarn deposits, which has guiding significance for identifying different types of skarn deposits.

## 7. Conclusions

(1) The Da'anhe deposit is a calcareous reduced skarn Au deposit related to the metasomatism of gabbro-dioritic magma and marble in the middle-deep part of the crust. The

    metallogenic process involved skarn and quartz-sulfide periods. Gold precipitated mainly in the late quartz sulfide stage and to a lesser extent in.

(2)    The initial ore-forming fluid of the Da'anhe deposit was derived from the gabbro-dioritic magma. Boiling and mixing were the main mechanisms driving gold and sulfide precipitation. The ore-forming material was mainly derived from the gabbro-dioritic magma.

(3)    The Da'anhe deposit was formed at depths ranging from 2.27 to 3.11 km, and is a medium to deep skarn gold deposit. Combined with regional observations, the rocks in the study area have been uplifted to the shallow surface (<500 m), and its prospecting potential is limited.

(4)    The distinctive nature of the ore-forming magma (source, reducing conditions and high water content, $\Delta$FMQ = 0.17, $\log(fO_2)$ = −15.11, water content = 6.80 wt.%) was key to the formation of the Da'anhe skarn gold deposit.

**Author Contributions:** Conceptualization, C.Z., F.S., J.W. and J.H.; Data curation, J.H., X.C., C.B., Z.X., L.Y. and S.H.; Formal analysis, J.S., J.W., D.Y., Z.X. and L.Y.; Funding acquisition, J.S. and J.W.; Investigation, C.Z., F.S., J.S., X.C. and C.B.; Methodology, C.Z., J.S., J.W. and X.C.; Project administration, C.Z., F.S., J.H. and S.H.; Resources, C.Z., F.S., J.S., J.W. and X.C.; Software, J.H., D.Y. and L.Y.; Supervision, C.Z., F.S., J.W., C.B. and Z.X.; Validation, C.B.; Visualization, J.S., J.H. and D.Y.; Writing—original draft, C.Z., F.S., J.S. and J.W. All authors have read and agreed to the published version of the manuscript.

**Funding:** This work was supported by the Second Tibetan Plateau Scientific Expedition and Research Program (STEP) (2022QZKK0201); the National Natural Science Foundation of China (Grant No. U20A2088; 42202068); the Kunlun Talented People of Qinghai Province, High-end Innovation and Entrepreneurship talents (Grant to Zhao Chuntao; E140DZ3901); the Special Research Assistant of Chinese Academy of Sciences (Grant to Zhao Chuntao); the Natural Science Foundation of Qinghai Province (No. 2020-ZJ-939Q); the Chinese Academy of Sciences Youth Innovation Promotion Association (No. 2023453).

**Data Availability Statement:** Data are contained within the article.

**Acknowledgments:** We sincerely thank guest editors Zhonghai Zhao, Jiafu Chen and Ju Nan for their efforts on this issue. Sincere thanks are extended to the three anonymous reviewers for their constructive comments.

**Conflicts of Interest:** No conflicts of interest exist in the submission of this manuscript, and the manuscript is approved by all authors for publication. I would like to declare on behalf of my coauthors that the work described was original research that has not been published previously, and is not under consideration for publication elsewhere, in whole or in part. All the listed authors have approved the manuscript that is enclosed.

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
