# Peer review of "Genesis and Prospecting Potential of the Da’anhe Skarn Au Deposit in the Central of the Lesser Xing’an Range, NE China: Evidence from Skarn Mineralogy, Fluid Inclusions and H-O Isotopes"

_minerals, doi:10.3390/min14030214_

Round 1
Reviewer 1 Report (Previous Reviewer 3)
Comments and Suggestions for Authors
The manuscript was improved significantly but there are still minor corrections necessary.
The abstract starts with a sentence without predicate: "Skarn Au deposits in the circum-Paleo-Pacific metallogenic belt." Please complete it!
"We have modified the labeling method for longitude and latitude. (line 102 in the revised manuscript)" - I think the suggestion was not understood correctly. I meant that the numbering on the edge of the map should indicate that longitudes are east from the prime meridian (like "130°E") and latitudes are north from the Equator (like "48°N").
Row 173: "crystallinities" - what does this mean? Grain size differences?
Figure 5: This is still not labelled correctly, although B-E seem to be TL xN, (transmitted light with crossed polarizers), A TL 1N, G-L RL 1N (reflected light with single polarizer), and F RL xN. For F and K scale is missing, in other cases the numbers are of different size, which makes an inattentive impression.
Figure 6 A-B: Lack of labelling like on Fig. 5.
Table 1: "(wt.%)" is true for the rows down to "Total" only. Below that it must be APFU down to Ca and maybe wt% again for the endmember components. Please make a correct distinction and captions! Maybe it would be better to divide the Table 1 into 1a, 1b and 1c parts.
Table 2 should be treated like Table 1.
Row 309: "were observed by petrography" - Petrography is not an observation method. You mean petrographic microscope, I think.
Table 3: "V/(V+L)%" - there isn't such a dimension. It must be volume concentration, V/V%.
Comments on the Quality of English LanguagePlease check also the typographic errors!
Author Response
Please see the attachment.

Reviewer 2 Report (New Reviewer)
Comments and Suggestions for Authors
This MS has undergone a round of review, and the quality of the article has been somewhat improved. However, there are still some flaws in the MS.
1> Some professional words are not used accurately, e.g., circum-Paleo-pacific metallogenic belt, the “circum-pacific metallogenic” is more appropriate. Do the authors confirm that diorite belong to mafic rock? (lines 53 to 54)
2> The authors should add the discussion about the relationships of the gabbro-diorite body age and the gold mineralization age.
3> Lines 694 to 695, “The initial ore-forming fluid of the Da'anhe deposit was derived from the fluid formed by the metasomatism of gabbro-dioritic magma”. This is a wrong conclusion, and the metasomatism itself can not produce fluid, the fluid only participates in the metasomatic reaction and acts as a medium.
4> The quality of MS English language is not good enough to be published, extensive editing of English language is required.
Comments on the Quality of English LanguageThe quality of MS English language is not good enough to be published, extensive editing of English language is required.
Author Response
Please see the attachment.

This manuscript is a resubmission of an earlier submission. The following is a list of the peer review reports and author responses from that submission.
Round 1
Reviewer 1 Report
Comments and Suggestions for Authors
Review on Minerals-2683085-peer-review-v1.pdf
The ms under consideration is dedicated to the possible mechanism of formation of the Da’anhe calcareous reduced skarn Au deposit. Text of the ms comprises valuable information about scarn mineralogy, fluid inclusions content and H-O isotope data. The ms should be of sure interest for a wide range of geologists and geochemists. My main objections concern to the lack of quantitative arguments used for description of the ore formation processes, thus a lot of authors’ statements look unfounded. I believe that thermodynamic justification of a set of processes should significantly improve the study.
As an example of the poorly justified statement - lines 620- 626:
“In the quartz-carbonate stage (stage II3), after boiling, mixing and water‒rock reactions of the initial ore-bearing hydrothermal solution, the metallogenic system have changed to low-temperature and low-salinity system (T = 137~247°C; S = 0.87~3.37% NaCl equiv.). At this time, meteoric water entered and dominated the metallogenic system, and large amounts of quartz and calcite began to precipitate from the fluid, forming quartz-carbonate veins and cutting through skarns and orebodies, marking the end of mineralization. “
What are the arguments for boiling? What is the content of the “initial ore-bearing hydrothermal solution”? What are the reasons for “and large amounts of quartz and calcite began to precipitate from the fluid”? Please provide quantitative substantiation for these statements.
Lines 660-663: The Da'anhe deposit is a calcareous reduced skarn Au deposit related to the metasomatism of gabbro-dioritic magma and marble in the middle-deep part of the crust. The metallogenic process experienced skarn period and quartz-sulfide period. Gold was mainly formed in the late quartz-sulfide stage, followed by the early quartz-sulfide stage.
Could you please propose a quantitative thermodynamic description of the stages?
Lines 664-666: The initial ore-forming fluid of the Da'anhe deposit was derived from the fluid formed by the metasomatism of gabbro-dioritic magma. Boiling and mixing are the main mechanisms of gold precipitation and mineralization.
What was the composition of «the initial ore forming fluid”? Please also argue why “boiling and mixing are the main mechanisms of gold precipitation and mineralization”.
Lines 669-671: “the contact zone between gabbroic diorite and marble in the northeastern part of the mining area has the potential for skarn gold deposits…”
Why? What is the physical-chemical reasoning for the Au deposition?
Lines 673 – 675: “The nature of the ore-forming magma (source, reduction and high water content) …is the key to the formation of the Da’anhe deposit.”
Really? What are the arguments? Is really the reduced fluid should be rich with Au? What are the main physical-chemical factors responsible for Au deposition?

Reviewer 2 Report
Comments and Suggestions for Authors
This article is well written. I just have one small question.
By comparing the S isotope range of pyrrhotite and pyrite in the Da'anhe deposit with that of skarn, the authors conclude that it is the source of magma. I think the conclusion is somewhat subjective and the evidence is not sufficient. I hope the author can provide the S-isotope results of magma and strata in the study area for comparison.
Reviewer 3 Report
Comments and Suggestions for Authors
The paper introduces interesting new results but must be improved both in formal and in factual aspects to be a worthy contribution to the knowledge on this deposit. The paragenetic model (emphasized also in the abstract as a new result) is not supported by documented observations; this definitely must be complemented.
Detailed remarks for the text:
Rows 127-128: So these "faults" are not faults but shear zones with both ductile and brittle deformation features. When You write "shear zones" and "schistosity zones", do you mean mylonite?
Row 141: "orebodies are controlled by the structure" - Please be more explicit, which structural elements? The shear zones? Folds?
Paragraph on page 7, rows 174-196: If the interpreted paragenetic sequence presented here is documented in another publication (even if not in English), please refer to this! If this is the result of the recent study, then it must be placed into the results chapter and it requires a detailed description of the observations used for this including sampling strategy, test methods and observed mineral associations! Are these indeed subsequent stages or zones rather? This should be discussed in the text.
Chapter 4: "Sampling" is in the title but its introduction is completely missing. Please extend the chapter corresponding to the title! Sample identifiers shown in the following tables are useless for the reader unless these can be located and linked to various rock types and parts of the mineralized rock body!
Rows 263-267, 280-286: It is superfluous to list data in sentences which are given in tables (instead of these, calculated parameters could stand in this form if appropriate, like mean, standard deviation etc); the relevant information is that all garnets belong to "grandite" and pyroxenes to di-hd series.
Row 480: It is better to name the "other researchers" as Yang, Sun and Fu and put the reference link here.
Tables 1,2: It would be better not to mix up chemical (oxide) composition with other things, which are not even explained in the caption!
Chapter 6.4: According to these data, mineralization "stages" seem to be rather depth zones! Do the calculated values correspond to the relative depth of the samples from which these were obtained?
Figures:
Figure 1: Latitutdes and longitudes should be indicated as North and East.
Figure 2: The legends should be reordered as usual (top-down, from youngest to oldest). Ages should be indicated for the sedimentary and volcanic rocks too. The word formation has to be capitalized in stratigraphic names. "Fracture" is shear zone, obviously.
Figure 3: It would be great to use the same symbols for the same units but not to use the same symbols for different units as on Figure 2. The legends could be then unified and not repeated (except additions here like alteration zones). "Diorite vein" - do you mean dyke? "Exploration line" - do you mean profile trace of fig B?
Figures 4-5: Explanation of the pictures is missing completely! (Rock types, sample locations/identifiers, transmitted/reflected light, 1N/xN etc.)
Figure 6: Are the stars representing the data taken from tables 1-2? (And Fig. 10 from table 4, 11-5, 12-6?) It should be referred.
Figure 11-12: If the data plotted here are the same as in Tables 5-6, the same reference should be given here!
Figure 13: This diagram should be reconsidered as it seems not to be consistent either with the body text or with Figure 3. Marble and metasandstone were described as rocks of the same formation before, yet here they are shown contacted with an angular unconformity? What is this "Si-Cal alteration"? What is the relationship of the "faults" to the intrusion? They seem to displace nothing but control the geometry of the ore bodies, although nothing was mentioned about this.
Comments on the Quality of English LanguageAt several points the text is not professional and sometimes it cannot be understood clearly. I do not give an extensive list of errors but indicate the most important problems (and corrections if it seems to be obvious).
Row 110: "metamorphic (quartz) sandstone" - This is not a proper rock name; metasandstone or quartzite?
Row 115: "are [...] unconformable on" - overlie unconformly
Row 122 (and several next occurrences): "skarnization" - There isn't such a term in proper English (see e.g. the Glossary of Geology). Skarn is not the name of a mineral group like sericite in sericitization etc. This metasomatic alteration is named as skarn formation in the textbooks.
The sentence here is supposed to mean that the host rock consists of a metasomatic mineral assemblage interpreted as endoskarn.
Rows 130-133: The sentences are supposed to mean that the mining area and the deposit is located on the SE limb of the NE striking Sanjiaoshan syncline within the Shenshu anticlinorium.
Rows 141-142: The ore bodies are NE or NW striking lenses or veins dipping steeply between 42-87 degrees.
Row 158: "developed" - of high grade? widely distributed? a general feature?
Rows 174-176: These are not structures, but form, as the ore appears: the ore occurs such as...
Row 203: "ground" - used?
Rows 258-259: Do you mean here that garnet is so rare that only one garnet grain was found and tested in your sample set? Or that all grains belong to the same garnet species?
Row 260 and next occurrences: "symbiotic" - associated with? (These are no living beings...)
Row 262: "has irregular grains" - the grains are anhedral?
Row 266: "The endmembers are composed of..." - No, the composition of the actual grains is expressed as percentage of the endmember components here. "Grossularite" is correctly named as grossular.
Row 275: "metasomatized by" - replaced by
Row 276: "relatively simple" - ??
Row 279: "particle sizes are generally 0.5~2 mm under polarization microscope" - Does the grain size depend on the device you observe it with?!
Rows 542-543: "...changed from hydrostatic pressure system to hydrostatic pressure system" - So there was no change?
Row 566: "ductile characteristics" - Ductile is an adjective of deformation and not of any rocks. There were no deformation features described previously. Did you mean simply that you assume no fracturing (and, in consequence, low permeability) in the country rocks?
Row 663: "the late quartz-sulfide stage, followed by the early quartz-sulfide stage" - obviously the sequence is the opposite.